# Companion cells with high florigen production express other small proteins and reveal a nitrogen-sensitive *FT* repressor

**Hiroshi Takagi**[1,2,3,4], **Shogo Ito**[1,5], **Jae Sung Shim**[1,6], **Akane Kubota**[1,7], **Andrew K Hempton**[1], **Nayoung Lee**[1,8], **Takamasa Suzuki**[9], **Jared S Wong**[1], **Chansie Yang**[1], **Christine T Nolan**[1], **Kerry L Bubb**[10], **Cristina M Alexandre**[10], **Daisuke Kurihara**[4,11], **Yoshikatsu Sato**[11], **Yasuomi Tada**[2,12], **Takatoshi Kiba**[13,14], **Jose L Pruneda-Paz**[15,16], **Christine Quietsch**[10,17], **Josh T Cuperus**[10,17], **Takato Imaizumi**[1]*

[1]Department of Biology, University of Washington, Seattle, United States; [2]Center for Gene Research, Nagoya University, Nagoya, Japan; [3]Bioscience and Biotechnology Center, Nagoya University, Nagoya, Japan; [4]Institute for Advanced Research (IAR), Nagoya University, Nagoya, Japan; [5]Department of Botany, Graduate School of Science, Kyoto University, Kyoto, Japan; [6]School of Biological Sciences and Technology, Chonnam National University, Gwangju, Republic of Korea; [7]Division of Biological Science, Nara Institute of Science and Technology, Nara, Japan; [8]Research Institute of Molecular Alchemy (RIMA), Gyeongsang National University, Jinju, Republic of Korea; [9]Department of Biological Chemistry, College of Bioscience and Biotechnology, Chubu University, Kasugai, Japan; [10]Department of Genome Sciences, University of Washington, Seattle, United States; [11]Institute of Transformative Bio-Molecules (WPI-ITbM), Nagoya University, Nagoya, Japan; [12]Division of Biological Science, Graduate School of Science, Nagoya University, Nagoya, Japan; [13]Graduate School of Bioagricultural Sciences, Nagoya University, Nagoya, Japan; [14]RIKEN Center for Sustainable Resource Science, Yokohama, Japan; [15]School of Biological Sciences, University of California San Diego, La Jolla, United States; [16]Center for Circadian Biology, University of California San Diego, La Jolla, United States; [17]Brotman Baty Institute for Precision Medicine, University of Washington, Seattle, United States

**\*For correspondence:**
takato@uw.edu

**Competing interest:** The authors declare that no competing interests exist.

## eLife Assessment

This **fundamental** study uncovers the unique molecular features of Arabidopsis phloem companion cells that highly express *FLOWERING LOCUS T* (*FT*). These *FT*-expressing cells constitute a distinct subpopulation marked by elevated ATP biosynthesis and co-expression of small mobile proteins such as FLP1 and BFT, highlighting a fine balance between florigen and anti-florigen signals. Motif analyses and transgenic studies further identify NIGT1 transcription factors as direct, nitrogen-inducible repressors of *FT*, providing a mechanism for delayed flowering under nitrogen-rich conditions. Together, the **compelling** findings show that florigen-producing companion cells integrate energy metabolism, systemic protein signals, and nutrient-responsive repression to fine-tune the seasonal and nutritional regulation of flowering.

**Abstract** The precise onset of flowering is crucial for successful reproduction. In longer days, the florigen gene *FLOWERING LOCUS T* (*FT*) is induced in specific leaf phloem companion cells in *Arabidopsis*. However, the molecular nature of these cells remains elusive. Here, we utilized bulk nuclei RNA-seq and single nuclei RNA (snRNA)-seq to investigate transcription in *FT*-expressing cells and other companion cells. Our bulk nuclei RNA-seq demonstrated that *FT*-expressing cells in cotyledons and true leaves showed differences in *FT* repressor gene expression. Within true leaves, our snRNA-seq analysis revealed that companion cells with high *FT* expression form a unique cluster. The cluster expresses other genes encoding small proteins, including the flowering and stem growth inducer FPF1-LIKE PROTEIN 1 (FLP1) and the anti-florigen BROTHER OF FT AND TFL1 (BFT). We also found that the promoters of *FT* and the genes co-expressed with *FT* in the cluster were enriched for the binding motif of NITRATE-INDUCIBLE GARP-TYPE TRANSCRIP-TIONAL REPRESSOR 1 (NIGT1). Overexpression of *NIGT1.2* and *NIGT1.4* repressed *FT* and delayed flowering under nitrogen-rich conditions, implying the roles of NIGT1s as nitrogen-dependent *FT* repressors. Taken together, our results indicate that unique *FT*-expressing phloem cells may produce multiple systemic signals to regulate plant growth and development.

## Introduction

Plants determine the seasonal timing of flowering based on environmental cues such as day length and temperature (*Andrés and Coupland, 2012*; *Song et al., 2015*; *Zhu et al., 2021*; *Freytes et al., 2021*; *Pandey et al., 2021*; *Takagi et al., 2023*). The small protein FLOWERING LOCUS T (FT) is a florigen, a mobile signaling molecule that promotes flowering (*Corbesier et al., 2007*; *Tamaki et al., 2007*; *Mathieu et al., 2007*; *Notaguchi et al., 2008*). In *Arabidopsis thaliana*, some phloem companion cells residing in the distal parts of leaves highly express *FT* (*Takada and Goto, 2003*; *Chen et al., 2018*). Although *FT* exhibits only a single expression peak in the evening under common laboratory long-day (LD) conditions, *FT* is highly expressed in the morning and evening under natural light conditions (*Song et al., 2018*; *Lee et al., 2023*). The discrepancy between natural and laboratory conditions can be attributed to differences in the red-to-far-red light (R/FR) ratios. Adjusting the R/FR ratio to that observed in nature is sufficient to recreate the bimodal expression pattern of *FT* in the lab (*Song et al., 2018*; *Lee et al., 2023*).

Despite extensive research on the genetic regulation of *FT* expression and function, our understanding of the cells that express *FT* is limited (*Takagi et al., 2023*). Although *FT* is expressed in some phloem companion cells, the precise molecular features that distinguish *FT*-expressing phloem companion cells from other companion cells have remained elusive (*Takagi et al., 2023*). Recently, to understand the distinct features of *FT*-expressing cells, we employed Translating Ribosome Affinity Purification (TRAP)-seq, a method that identifies ribosome-associated transcripts in specific tissues or cell types (*Takagi et al., 2025*). We found that the list of differentially translated transcripts in *FT*-expressing cells partially overlapped with that in the general phloem companion cell marker gene *SUCROSE-PROTON SYMPORTER 2* (*SUC2*)-expressing cells but also contained unique transcripts to *FT*-expressing cells (*Takagi et al., 2025*). This supports the notion that the *FT*-expressing cells are similar but different from general phloem companion cells (*Takada and Goto, 2003*; *Chen et al., 2018*). Our TRAP-seq datasets showed that several *FT* positive and negative transcriptional regulators and proteins involved in FT protein transport were specifically enriched in *FT*-expressing cells. We also found that a gene encoding the small protein FPF1-LIKE PROTEIN 1 (FLP1) is highly expressed in *FT*-expressing cells under LD conditions with adjusted R/FR ratio (hereafter, LD +FR conditions), but not under standard laboratory LD conditions (*Takagi et al., 2025*). Further analysis revealed that FLP1 promotes flowering and initial inflorescence stem growth, suggesting that it orchestrates flowering and inflorescence stem growth during the transition from the vegetative to the reproductive stage.

Although our bulk tissue/cell-specific TRAP-seq analysis revealed some unique characteristics of *FT*-expressing cells, we still do not know their characteristics in a single-cell resolution. As *FT* is expressed at specific times of the day in relatively small numbers of phloem companion cells, we needed a method to collect the information from these rare *FT*-expressing cells in a time-dependent manner. Single-cell RNA-seq (scRNA-seq) is a powerful tool to study different cell populations within the complexity of biological tissues. In plant studies, especially for root tissues, the most popular procedure involves protoplasting of cells and the application of Drop-seq in the 10X Genomics

pipeline. However, protoplasting is not ideal for isolating highly embedded phloem companion cells efficiently (*Kim et al., 2021*). On top of this problem, our target is *FT*-expressing companion cells at the specific time point of the *FT* morning peak, thus the time required for tissue isolation must be short, so that the isolated cells still restore the time-of-day information. To overcome these technical challenges, we deployed single-nucleus RNA-seq (snRNA-seq) combined with fluorescence-activated nuclei sorting (FANS) to isolate GFP-labeled cell-type-specific nuclei at a specific time of the day. Here, we report the unique characteristics of *FT*-expressing cells at a single-cell level and identify new, growth condition-specific transcriptional repressors of *FT*.

## Results
### Bulk nuclei RNA-seq analysis finds profound expression differences in *FT*-expressing cells between cotyledons and true leaves

To investigate gene expression in the nuclei of specific cell populations, we used *Arabidopsis* transgenic lines expressing the gene encoding Nuclear Targeting Fusion (NTF) protein consists of the nuclear envelope-targeting WPP domain, green fluorescent protein (GFP), and biotin ligase recognition peptide (BLRP; *Deal and Henikoff, 2011*) under the control of cell-type-specific promoters (*Figure 1—figure supplement 1A*). In addition to the previously generated *pFT:NTF* line (*Takagi et al., 2025*), we generated transgenic lines with *NTF* controlled by the promoter of *SUC2* (*pSUC2*) to capture most phloem companion cells, the promoter of *CHLOROPHYLL A/B-BINDING PROTEIN 2* (*pCAB2*) for mesophyll cells, and *p35S* promoter as a non-specific control (*Figure 1A*, *Figure 1—figure supplement 1A*). Although our original intention was the utilization of the INTACT (isolation of nuclei tagged in specific cell types), the method to capture tissue-specifically biotinylated nuclei using magnetic beads (*Deal and Henikoff, 2011*), we instead employed FANS to isolate the respective GFP-positive nuclei (*Shi et al., 2021*).

Although *FT* is expressed in both cotyledons and true leaves, it was heretofore unknown whether *FT*-expressing cells in cotyledons and true leaves were equivalent to one another at the whole transcriptome level or showed tissue-specific differences in transcription. The previous histological analyses of the *Arabidopsis pFT:GUS* reporter line showed that *FT* expression in true leaves was confined to the distal part of the leaf vasculature, whereas in cotyledons, *FT* expression was observed in the broader part of the vein (*Takada and Goto, 2003*; *Ito et al., 2012*), suggesting different profiles of *FT*-expressing cells in cotyledons and true leaves. To exclude possible tissue-specificity as a source of noise in our single-nucleus data, we conducted bulk nuclear RNA-seq of GFP-positive nuclei isolated from either cotyledons or true leaves. To do so, we harvested cotyledons and the first set of two true leaves from 2-week-old transgenic plants grown under LD +FR conditions at Zeitgeber time 4 (ZT4), the time of the *FT* morning expression peak, and isolated GFP-positive nuclei within an hour using a cell sorter (*Figure 1—figure supplement 1B and D*). We successfully collected GFP-positive nuclei from both cotyledons and true leaves for all transgenic lines (*Figure 1—figure supplement 2*) and conducted bulk RNA-seq.

The principal component analysis (PCA) of the resulting expression data indicated separation by tissue and targeted cell population (*Figure 1B*). Interestingly, the biggest difference among these datasets was the difference between cotyledons and true leaves, rather than a cell/tissue-type difference. Among different cells/tissues, *FT*-expressing cells show a distinct gene expression profile compared with other cell/tissue types examined.

Next, to ensure that we had properly enriched for the targeted cell populations, we conducted pairwise comparisons between the *p35S:NTF* control line and the cell type-specific *NTF* lines (*Figure 1—source data 1–6*). As expected, in true leaves of the *pSUC2:NTF* line, phloem companion cell marker genes *FT*, *SUC2*, *AHA3*, *APL*, *CM3*, and *AAP4* showed higher expression than in the *p35S:NTF* line, and the *pFT:NTF* line showed strong upregulation of *FT* in both cotyledon and true leaves (*Figure 1C*, *Figure 1—figure supplement 3A-E*). True leaves of the *pFT:NTF* line also showed upregulation of other phloem companion cell marker genes such as *AHA3*, *APL*, and *AAP4*; however, cotyledons of the *pFT:NTF* line only showed upregulation of *AAP4* (*Figure 1—figure supplement 3A-E*), indicating some of the known vascular marker genes are more suitable for representing vascular tissues in true leaves.

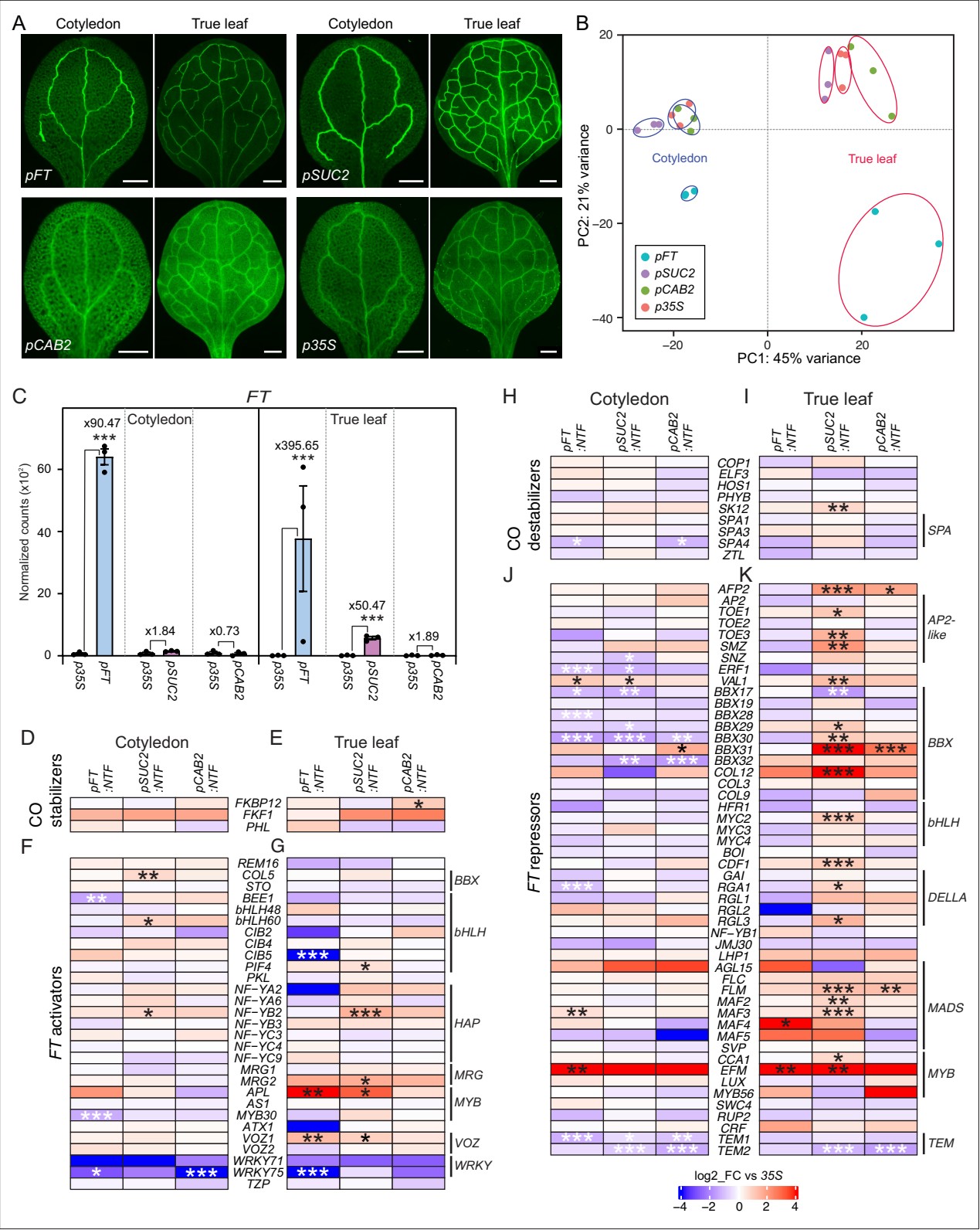

**Figure 1.** Tissue- and cell-type-specific gene expression in cotyledons and true leaves. (**A**) Representative GFP fluorescence images of ClearSee-treated cotyledons and true leaves of *pFT:NTF*, *pSUC2:NTF*, *pCAB2:NTF*, and *p35:NTF* lines. Scale bar, 500 µm. (**B**) The first two principal components of bulk RNA-seq analysis for three independent cotyledon and true leaf samples. Cotyledon and true leaf samples are circled in blue and red, respectively. (**C**) DEseq2-normalized counts of *FT* transcripts in sorted nuclei for the *pFT:NFT*, *pSUC2:NFT*, and *pCAB2:NFT* lines compared to the *p35S:NTF* line.

*Figure 1 continued on next page*

*Figure 1 continued*

Fold enrichments of *FT* transcripts in each *NTF* line compared with those in *p35S:NTF* line are indicated (n=3). \*\*\*padj <0.001. (**D–K**) Expression of genes encoding CO stabilizers (**D and E**), *FT* activators (**F and G**), CO destabilizers (**H and I**), and *FT* repressors (**J and K**) in cotyledons (**D, F, H, and J**) and true leaves (**E, G, I, and K**). Genes encoding proteins belonging to the same family were clustered in the heatmap. Bar color indicates log2-scaled fold-change relative to the *p35S:NTF* line. Asterisks denote significant differences from *p35S:NTF* (\*padj <0.05; \*\*padj <0.01; \*\*\*padj <0.001). CO was removed from the analysis due to insufficient reads at ZT4.

The online version of this article includes the following source data and figure supplement(s) for figure 1:

**Source data 1.** A list of genes expressed in cotyledons of pFT:NTF plants compared with those in cotyledons of p35S:NTF plants (n=3).

**Source data 2.** A list of genes expressed in cotyledons of pSUC2:NTF plants compared with those in cotyledons of p35S:NTF plants (n=3).

**Source data 3.** A list of genes expressed in cotyledons of pCAB2:NTF plants compared with those in cotyledons of p35S:NTF plants (n=3).

**Source data 4.** A list of genes expressed in true leaves of pFT:NTF plants compared with those in true leaves of p35S:NTF plants (n=3).

**Source data 5.** A list of genes expressed in true leaves of pSUC2:NTF plants compared with those in true leaves of p35S:NTF plants (n=3).

**Source data 6.** A list of genes expressed in true leaves of pCAB2:NTF plants compared with those in true leaves of p35S:NTF plants (n=3).

**Source data 7.** Known spatial expression patterns of genes whose expression areis higher in pSUC2:NTF cotyledons than in p35S:NTF cotyledons.

**Figure supplement 1.** A diagram of constructs that express tissue/cell-specific *NTF* genes and flowcharts of bulk and single nuclei RNA-seq procedures.

**Figure supplement 2.** Plots showing 100,000 events of fluorescence-activated nuclei sorting (FANS).

**Figure supplement 3.** DEseq2-normalized expression of phloem companion cell marker genes (**A–E**), mesophyll cell marker genes (**F and G**), and FT transporting genes (**H–I**) from sorted nuclei bulk RNA-seq of cotyledons and true leaves in the *pFT:NTF* (shown as *pFT*), *pSUC2:NTF* (*pSUC2*), *pCAB2:NTF* (*pCAB2*), and *p35S:NTF* (*p35S*) plants.

**Figure supplement 4.** Cotyledons and true leaves express unique sets of upregulated genes in the sorted nuclei.

**Figure supplement 5.** Cotyledons and true leaves express unique sets of downregulated genes in the sorted nuclei.

Similarly, in the *pSUC2:NTF* line, we observed differences in cell marker gene expression between cotyledons and true leaves. In cotyledons of the *pSUC2:NTF* line, *CM3* and *AAP4* were upregulated, but not *SUC2*, *FT*, *AHA3*, and *APL*. To ensure that FANS properly enriched nuclei from *pSUC2:NTF* cotyledons, we checked the spatial expression pattern of each gene that was highly expressed in *pSUC2:NTF* cotyledons [adjusted *P*-value (*padj*) <0.05, fold-change >twofold, *Figure 1—source data 7*]. The majority of the genes upregulated in the cotyledons of our *pSUC2:NTF* line were found to be expressed in the vasculature. As expected, expression of mesophyll marker gene *CAB2* was significantly lower in *pSUC2:NTF* cotyledon samples (padj = $1.5 \times 10^{-4}$), in addition to the mesophyll cell marker *RBCS1A* in true leaf samples, suggesting that a significant amount of mesophyll cells was successfully removed by FANS (*Figure 1—figure supplement 3F and G*).

Having ensured proper enrichments of the targeted nuclei, we compared global changes in expression among the tested transgenic lines. Consistent with the PCA analysis, the vast majority of differentially expressed genes within a given transgenic line did not overlap between cotyledons and true leaves (*Figure 1—figure supplements 4 and 5*). Next, we compared the expression of known *FT* positive and negative regulators in cotyledons and true leaves of the *pFT:NTF*, *pSUC2:NTF*, and *pCAB2:NTF* lines (*Figure 1D–K*; *Takagi et al., 2023*).

Among the transcription factors that promote *FT* expression, *APL* and *VOZ1* were upregulated in true leaves of the *pFT:NTF* and *pSUC2:NTF* lines (*Figure 1G*). In true leaves of the *pSUC2:NTF* line, a CO destabilizing factor was significantly upregulated (*Figure 1I*), in addition to TFs that repress *FT* expression (*Figure 1K*). In contrast to *pSUC2:NTF* true leaves, true leaves of the *pFT:NTF* line did not show significant upregulation for most of these *FT* repressors. This lack of *FT* repression appears to be crucial for specifying *FT*-expressing cells. In contrast, there were far fewer expression differences in cotyledons between the *pFT:NTF* and *pSUC2:NTF* lines, indicating that the respective cells in true leaves have diverged more than comparable ones in cotyledons.

Similarly, *FT* expression and FT transport appeared to be more tightly associated in true leaves than in cotyledons. As reported, the expression of the FT transporter gene *FTIP1* is highly associated with *FT* expression (*Takagi et al., 2025*). Consistent with this earlier finding, among FT transporter genes, we observed significant upregulation of *FTIP1* expression in true leaves of the *pSUC2:NTF* and the *pFT:NTF* lines but not in the respective cotyledon samples (*Figure 1—figure supplement 3H-K*).

To further support true leaves as most suitable for investigating *FT*-expressing cells, we performed Terms enrichment analysis with Metascape (*Zhou et al., 2019*). In the *pFT:NTF* line, the genes involved in the 'regulation of reproductive process' were upregulated in both cotyledons and true leaves, while the genes related to the term 'long-day photoperiodism, flowering' were only upregulated in the true leaf samples of the *pFT:NTF* line (*Figure 1—figure supplement 4*).

## snRNA-seq revealed a subpopulation of phloem companion cells that highly express *FT*

Next, we used true leaves of the *pFT:NTF* and *pSUC2:NTF* lines for snRNA-seq using the 10X Genomics Chromium platform (*Figure 1—figure supplement 1C*). We captured a total of 1173 nuclei for the *pFT:NTF* line and 3650 nuclei for the *pSUC2:NTF* line. In total, we captured 4823 nuclei and detected 20,732 genes, a median of 149 genes per nuclei (*Figure 2A–C*). Gene expression of these nuclei was projected in a two-dimensional UMAP as 11 different clusters (*McInnes et al., 2018*; *Figure 2D*). The nuclei in clusters 8 and 10 showed substantially higher numbers of transcripts than other clusters (*Figure 2—figure supplement 1A and B*). Because this discrepancy made comparisons difficult, we focused on analyzing gene expression in the other clusters.

We first asked which nuclei highly expressed *FT*. These nuclei resided in cluster 7 (*Figure 2D and E*). *SUC2* expression showed a far broader distribution across clusters than *FT* expression (*Figure 2F*). The nuclei in cluster 7 significantly highly expressed 268 genes compared with the total population (*Figure 2—source data 1*). Among these genes, we found *FLP1* (*Figure 2G*), a gene encoding a flowering-promoting factor acting in parallel with FT (*Takagi et al., 2025*). Next, we cross-checked these 268 genes with genes identified in the previous TRAP-seq experiment (*Takagi et al., 2025*). 202 of the 268 genes showed higher expression in *FT*-expressing cells (*pFT:FLAG-GFP-RPL18*) than in whole companion cells (*pSUC2:FLAG-GFP-RPL18*) (*Figure 2—figure supplement 2A*) in the prior study, validating our single nuclei analysis. Further, *FKBP12*, a gene encoding a CO stabilizing protein, was expressed in cluster 7 (*Figure 2—figure supplement 2B*; *Serrano-Bueno et al., 2020*).

To explore the *FT*-expressing cluster 7, we conducted Terms enrichment analysis using Metascape (*Zhou et al., 2019*). Genes involved in 'Oxidative phosphorylation' were enriched in cluster 7 (*Figure 2—figure supplement 2C*), suggesting that ATP synthesis is particularly active in nuclei belonging to this cluster (*Figure 2H*). Genes involved in proton-generating complex I–IV and ATP synthesizing complex V in mitochondrion membrane were also upregulated in cluster 7 (*Figure 2—figure supplement 2D*), suggesting that the entire ATP synthesis pathway is activated in *FT*-expressing nuclei. Additional terms such as 'generation of precursor metabolite and energy' and 'purine nucleotide triphosphate metabolic process', further indicated upregulation of ATP production. Since phloem companion cells actively hydrolyze ATP to generate proton gradients to load sucrose and amino acids (*Hunt et al., 2023*), it is plausible that *FT*-expressing phloem companion cells generate a substantial amount of ATP for transport.

We next inquired about the characteristics of other phloem nuclei clusters. Like cluster 7, cluster 4 showed significantly higher *SUC2* expression compared with other clusters, suggesting companion cell identity (*Figure 2D and F*, *Figure 2—source data 1*). Moreover, the genes *LTP1*, *MLP28*, and *XTH4* were upregulated in cluster 4, consistent with prior studies showing their exclusive expression in vascular tissues (particularly in the main vein) that include phloem (*Figure 2—figure supplement 3A–C*; *Kushwah et al., 2020*; *Litholdo et al., 2016*; *Wang et al., 2016*). Terms enrichment analysis found that genes encoding aquaporin were enriched in cluster 4 (*Figure 2I*, *Figure 2—figure supplement 3D and E*). To understand which part of leaf tissue consists of cluster 4 cells, we marked the promoter activity of *PLASMA MEMBRANE INTRINSIC PROTEIN 2.6* (*PIP2;6*), an aquaporin gene highly expressed in cluster 4, using the GFP-fusion protein (*Figure 2—figure supplement 3F and G*). It showed the strongest signal in the main vein, consistent with the previous GUS reporter assay (*Prado et al., 2013*). Taken together, the nuclei in cluster 4 are likely to be derived from phloem cells in the main vein that are active in solute transport.

Cluster 5 is mostly composed of nuclei from the *pSUC2:NTF* line (464 out of 515 nuclei; *Figure 2—figure supplement 1B*). The nuclei in this cluster showed differentially expressed genes related to defense (*Figure 2—figure supplement 4A*). Specifically, *MYC2*, a master regulator of response to jasmonic acid (JA), and the JA biosynthetic genes *LOX3*, *LOX4*, *OPR3*, and *OPCL1* are expressed in cluster 5 (*Figure 2—figure supplement 4B*). These genes are known to be expressed in vascular

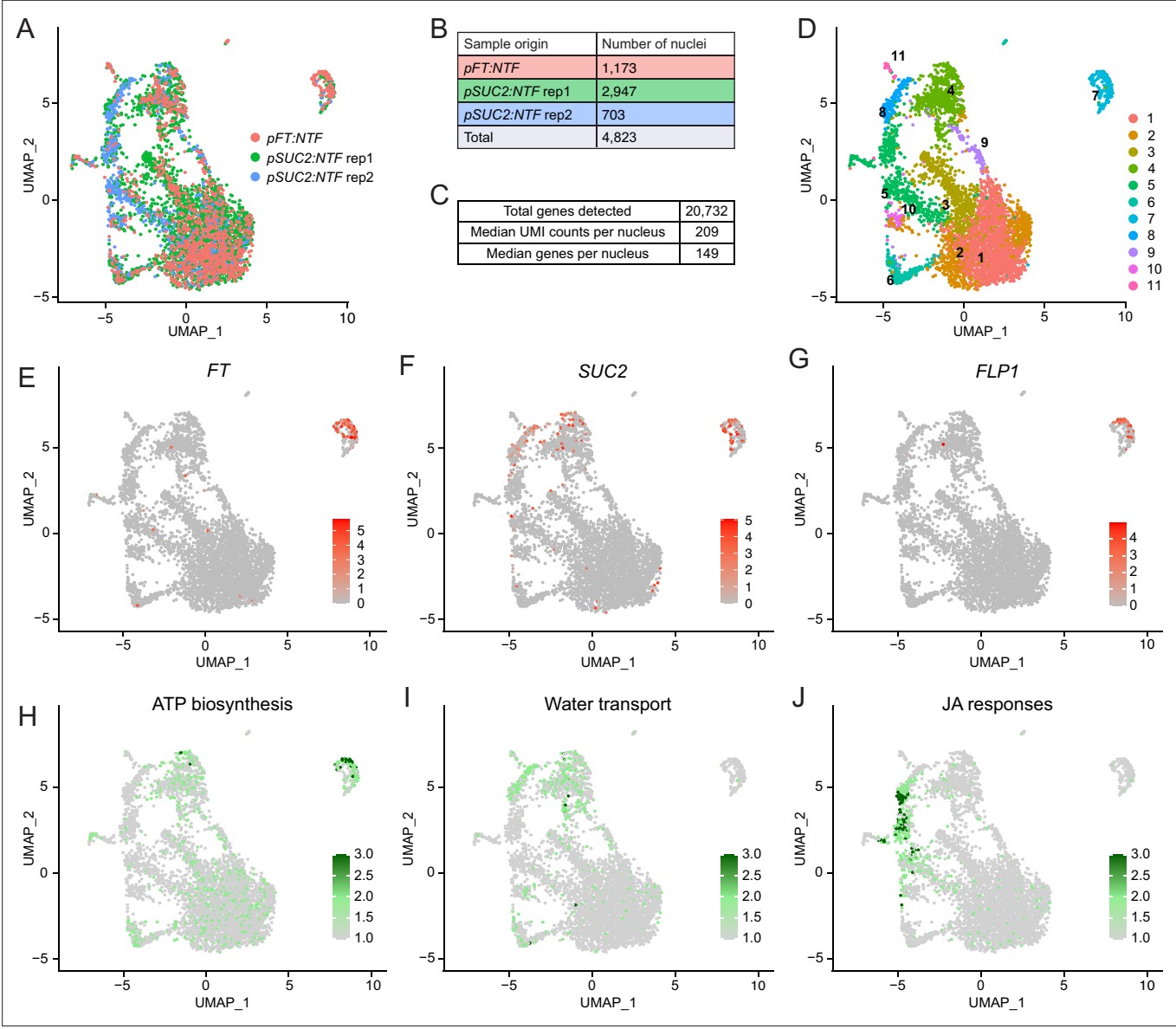

**Figure 2.** Single-nucleus RNA-seq identifies distinct subpopulations of phloem companion cells. (**A**) UMAP with the origin of nuclei indicated by color. (**B and C**) Tables with the number of nuclei originated from each line and sample (**B**) and median UMI and number of genes detected per nucleus (**C**). (**D**) Seurat defined clusters with cluster numbers indicated. (E–G) UMAP annotated with normalized read counts for *FT* (**E**), *SUC2* (**F**), and *FLP1* (**G**) expression. (H–I) UMAP annotated with average read counts of genes related to ATP biosynthesis (**H**), water transport (**I**), and JA responses (**J**). Color bars indicate gene expression levels. For the lists of genes for H–I, see *Figure 2—figure supplement 2D* (pink highlighted), 3E, and 4B, respectively.

The online version of this article includes the following source data and figure supplement(s) for figure 2:

**Source data 1.** Highly expressed genes in each cell cluster compared with the average cell population in the snRNA-seq analysis in *pFT:NTF and pSUC2:NTF* plant data combined.

**Figure supplement 1.** Characteristics of UMAP clusters 8 and 10.

**Figure supplement 2.** Characteristics of the *FT*-expressing UMAP cluster 7.

**Figure supplement 3.** Characteristics of UMAP cluster 4.

**Figure supplement 4.** Characteristics of UMAP cluster 5.

**Figure supplement 5.** Characteristics of UMAP cluster 6.

**Figure supplement 6.** Protoplast isolation procedure and the expression of tissue-specific marker genes in isolated protoplasts.

tissues, with *OPR3* expressed in phloem companion cells and *LOX3/4* in phloem (*Chauvin et al., 2016*; *Kienow et al., 2008*; *Li et al., 2013*). This subpopulation of phloem cells appears to play pivotal roles in JA-dependent defense responses. We did not include a nuclei fixation step in our FANS protocol, because we encountered tissue/nucleus clumping with fixation, which caused an issue in the sorting process. Because these cells were not fixed, the nuclei isolation process may have triggered wounding responses due to chopping samples before the sorting process (*Figure 1—figure supplement 1C*).

Cluster 6 consisted of both phloem parenchyma and bundle sheath cells (*Figure 2—figure supplement 5A and B*). Subclustering revealed that the nuclei in the upper part of cluster 6 expressed phloem parenchyma marker genes, and the nuclei in the lower part expressed bundle sheath marker genes (*Kim et al., 2021*; *Figure 2—figure supplement 5C–F*). Overall, the genes expressed in this cluster were enriched for genes functioning in sulfur metabolisms (*Figure 2—figure supplement 5G*), including glucosinolate biosynthesis (*Figure 2—figure supplement 5H*). A previous study showed that sulfur metabolic and glucosinolate biosynthetic genes are actively expressed in bundle sheath cells (*Aubry et al., 2014*).

Lastly, Clusters 1, 2, and 3 showed expression of mesophyll cell marker genes (*Figure 2—figure supplement 1C–F*), suggesting that some mesophyll cells were included during the sorting process. This result might be due to weak induction of the *FT* and *SUC2* promoters in mesophyll cells (*Takagi et al., 2025*). Indeed, we found both genes to be expressed at low levels in mesophyll protoplasts (*Figure 2—figure supplement 6*). In summary, using snRNA-seq combined with FANS, we revealed the presence of a unique subpopulation of cells with high nuclear *FT* expression. In these nuclei, genes related to ATP synthesis were enriched. Moreover, we identified other phloem cell clusters enriched for genes in water transport and JA response, which were not previously reported (*Kim et al., 2021*).

Next, we subclustered the nuclei in cluster 7, finding three subclusters (*Figure 3A and B*). Subcluster 7.2 contained the nuclei with the highest *FT* expression, in addition to expression of the companion cell marker genes, *SUC2* and *AHA3*. These nuclei contained fewer transcripts of the mesophyll marker genes *RBCS1A*, *CAB3*, and *CAB2* (*Figure 3C and D*), consistent with a prior scRNA-seq study showing the negative correlation between the expression of vasculature and mesophyll cell marker genes (*Torii et al., 2022*). The vascular tissue located at the distal part of true leaves, where *FT* is expressed highly, is developmentally old, whereas veins emerging from the bottom part of leaves are developmentally young (*Biedroń and Banasiak, 2018*). Thus, subcluster 7.3 may be comprised of nuclei from developmentally younger companion cells that have not fully matured yet, whereas those in subcluster 7.2 are older and have gained a stronger companion cell identity.

## Phloem companion cells with high *FT* expression express other genes encoding small proteins

In LD +FR conditions, *FT*-expressing cells also express *FLP1*, which encodes another small protein with systemic effects on flowering and inflorescent stem growth (*Takagi et al., 2025*). We asked whether the cluster 7 nuclei might express other genes encoding small proteins, which could move through phloem flow. Indeed, the median number of amino acids encoded in the differentially expressed genes of cluster 7 is smaller than those in clusters 4 and 5 (*Figure 4A*). To validate whether high *FT*-expressing vascular cells co-express genes specifically upregulated in cluster 7, which encode small proteins, we selected eight such genes and generated their respective promoter fusions with *H2B-tdTomato* in the *pFT:NTF* background. The spatial expression patterns of these eight genes and high *FT* signals overlapped (*Figure 4B*, *Figure 4—figure supplement 1*). *FT* and cluster 7-enriched genes examined tended to express highly in the distal part of minor veins, but weakly in the main vein adjacent to the petiole, showing a clear difference from the cluster 4-enriched gene *PIP2;6* that is expressed highly in the main vein (*Figure 2—figure supplement 3F*, *Figure 4—figure supplement 1*). It should also be noted that the spatial expression patterns of *FT* and other cluster 7-enriched genes did not completely overlap; there were cells expressing only one of them at high levels (*Figure 4—figure supplements 1 and 2*).

To assess the biological significance of *FT* generated from cluster 7, we knocked down *FT* transcripts using an artificial microRNA (amiRNA) targeted to *FT* (*amiR-ft*) controlled by the promoter of cluster 7-enriched *ROXY10* gene. As a comparison, we also expressed the *amiR-ft* gene using *SUC2* (general companion cell-marker gene), *PIP2;6* (cluster 4-enriched gene), and *GC1* a marker gene for guard cells where *FT* is also expressed (*Yang et al., 2008*; *Kinoshita et al., 2011*), and tested the

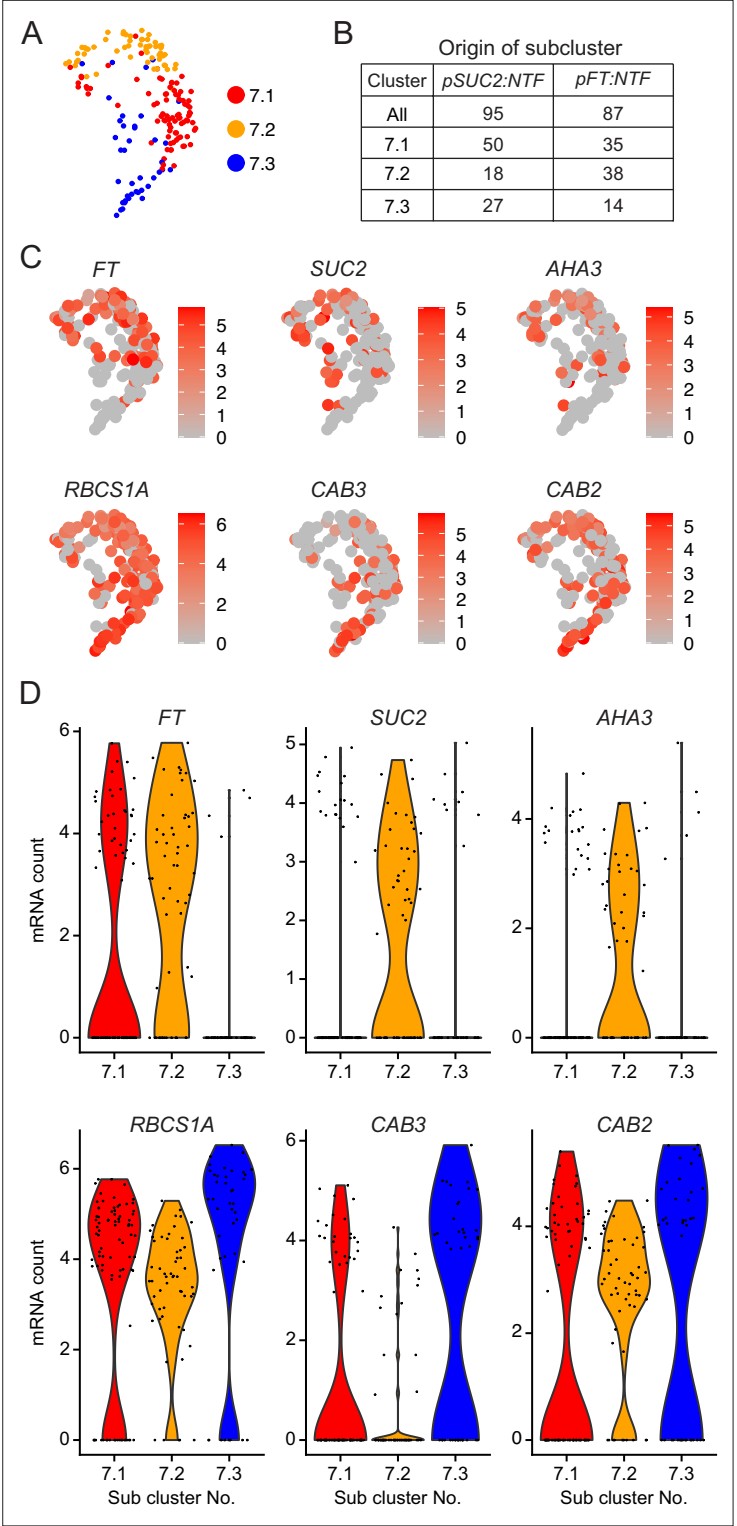

**Figure 3.** Phloem companion cell and mesophyll cell marker gene expression in cluster 7. (**A**) Subclustering of cluster 7. Colors indicate each subcluster. (**B**) Table showing the number of nuclei originated from each *NTF* line in cluster 7. (**C**) UMAP of normalized read counts of *FT* and companion cell marker genes (*SUC2*, and *AHA3*) and mesophyll cell marker genes (*RBCS1A*, *CAB3*, and *CAB2*). (**D**) Violin plot of normalized read counts of *FT* and marker genes for companion cells and mesophyll cells across the three subclusters.

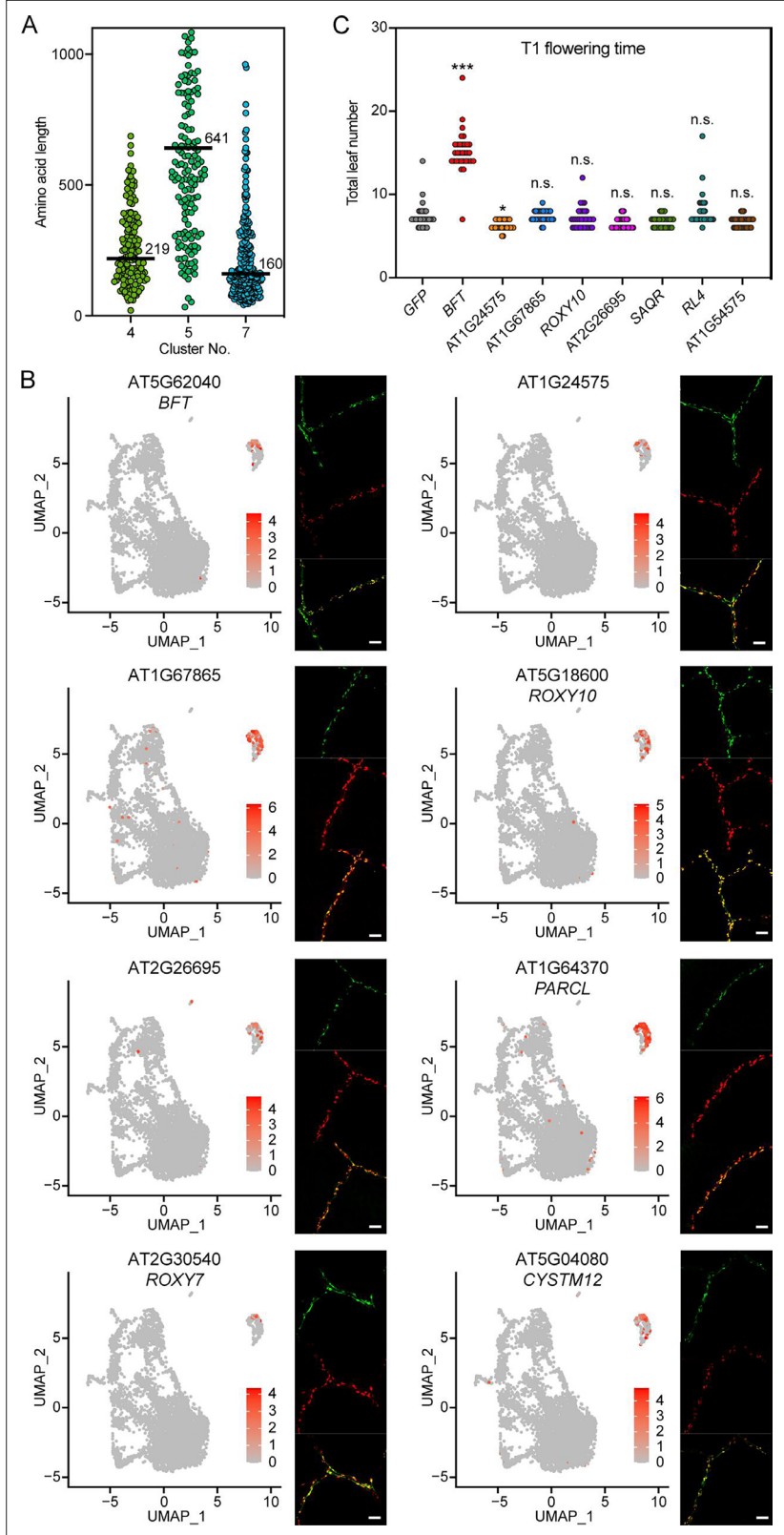

**Figure 4.** *FT*-expressing cells express genes encoding other small proteins. (**A**) Amino acid length of proteins encoded by genes differentially expressed in clusters 4, 5, and 7. Black bars indicate the median amino acid length. (**B**) Expression of the *pFT:NFT* line (green) and promoter fusions of selected cluster 7 genes with *H2B-tdTomato* (red) in true leaves. The selected genes were differentially expressed in cluster 7 and encoded small

*Figure 4 continued on next page*

*Figure 4 continued*

proteins. The yellow color shows an overlap between green and red signals. Scale bar, 50 μm. (**C**) Flowering time measurements of $T_1$ transgenic plants overexpressing selected cluster 7 genes driven by the *pSUC2* promoter (n≥31). Eight genes were tested, five of which were tested for overlap with *FT* expression in (**B**). Asterisks denote significant differences from *GFP* (*p<0.05, ***p<0.001, one-way ANOVA and Dunnett's multiple comparison test). n.s. indicates not significant.

The online version of this article includes the following figure supplement(s) for figure 4:

**Figure supplement 1.** Spatial expression patterns of GFP signal derived from the *pFT:NFT* (**A, D, and G**) and H2B-tdTomato signals controlled by the promoters of representatives of cluster 7-enriched genes: *ROXY10* (**B**), *BFT* (**E**), and *AT1G24574* (**H**) in true leaves.

**Figure supplement 2.** Enlarged images of phloem companion cells showing spatial expression overlap between *FT* and cluster 7 genes.

**Figure supplement 3.** The silencing of *FT* using amiRNA to confirm the overlap of spatial expression between *FT* and cluster 7-specific *ROXY10* genes.

effect on the timing of flowering at the $T_1$ generation. If *FT* generated from cluster 7 is important, as *ROXY10* is specifically expressed in cluster 7, late flowering of *pROXY10:amiRNA-ft* is anticipated. As a result, *pROXY10:amiRNA-ft* and *pSUC2:amiR-ft* lines severely delayed flowering (in fact, *pSUC2:amiR-NA-ft* flowered later than *ft-101* mutant, likely due to the effect of the hygromycin selection plate). On the other hand, *pPIP2;6:amiR-ft* was comparable with the negative control *pGC1:amiRNA-ft* line (*Figure 4—figure supplement 3A*). The basal expression level of *ROXY10* was much lower than that of *SUC2* and *PIP2;6* (*Figure 4—figure supplement 3B*); therefore, the delayed flowering phenotype of *pROXY10:amiR-ft* lines was not attributed to the promoter strength of genes used to drive the amiRNA. These results indicate that *FT* expressed in a relatively small but unique cluster 7 is biologically significant for flowering time regulation, although *FT* expressed in other companion cells other than cluster 7 cells may also contribute to flowering.

Next, we asked whether some of the genes enriched in cluster 7 encode new systemic flowering or growth regulators. We selected eight genes, five of which were tested for overlap with *FT* expression (*Figure 4B*) and overexpressed each one of them using the *SUC2* promoter. Some of these genes were previously implicated in flowering. *BFT* is an *FT* homolog and acts as a floral repressor. *SENESCENCE-ASSOCIATED AND QQS-RELATED* (*SAQR*) promotes flowering under short-day (SD) conditions (*Jones et al., 2016*). *RAD-LIKE 4* (*RL4*) has high homology to the *RADIALIS* (*RAD*) gene in *Antirrhinum majus*, whose ectopic expression in *Arabidopsis* causes growth defects and late flowering (*Baxter et al., 2007*). We measured the flowering time of the overexpression lines in independent $T_1$ plants (*Figure 4C*). $T_1$ *pSUC2:BFT* plants grown under LD +FR conditions showed severely delayed flowering compared with *pSUC2:GFP* control lines (*Figure 4C*). This result is in stark contrast to prior results obtained under standard laboratory light conditions (*Ryu et al., 2014*), suggesting that BFT proteins might be mobile or more functionally active under natural light conditions. None of the other transgenic lines showed noticeable changes in flowering time, although the leaf numbers slightly decreased in the AT1G24575 overexpressing lines.

## Motif enrichment analysis and transgenic studies identify NIGT1 transcription factors as direct *FT* repressors

We performed motif enrichment analysis on the promoters of the 268 genes differentially expressed in cluster 7 (*Dreos et al., 2015*; *Meylan et al., 2020*), using randomly selected 3000 genes as a control set. The most enriched motif was the motif of the NITRATE-INDUCIBLE GARP-TYPE TRANSCRIPTIONAL REPRESSOR 1.2 (also known as HHO2, $p=9.3 \times 10^{-6}$) (*Figure 5—source data 1*)(*Figure 5*). Although the NIGT1/HHO transcription factors (TFs) are primarily known as repressors of genes involved in nitrate uptake, constitutive expression of *NIGT1.2* significantly decreases *FT* expression (*Kiba et al., 2018*; *Maeda et al., 2018*). In fact, there are two potential NIGT1 binding sites located within and adjacent to the *shadow 1* and *2* domains (*S1 and S2*) in the *FT* regulatory region (*Adrian et al., 2010*; *Figure 5—figure supplement 1A*). These domains are highly conserved among *Brassicaceae* species and important for *FT* transcriptional regulation. Moreover, the NIGT1 binding motifs reside in close proximity to the CO binding motif (TGTGNNATG, CO-responsive element, CORE; *Adrian et al., 2010*; *Tiwari et al., 2010*), suggesting that NIGT1 binding could affect CO binding.

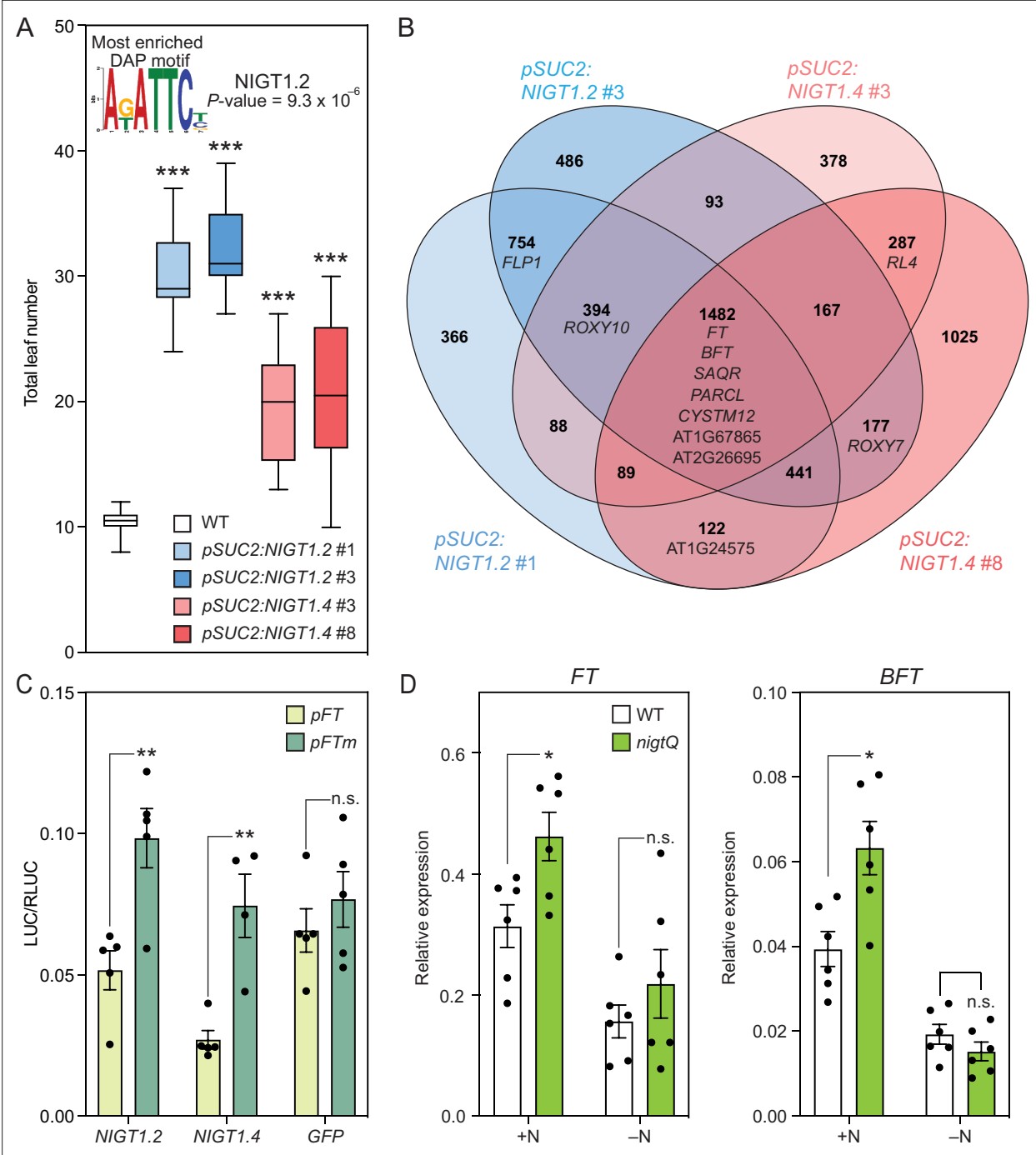

**Figure 5.** NIGT1 transcription factors are repressors of *FT*. (**A**) Motif enrichment analysis using the 500 bp promoters of 268 differentially expressed genes in cluster 7, and flowering time of the *pSUC2:NIGT1.2* and *pSUC2:NIGT1.4* lines in LD +FR. The NIGT1.2 binding site was the most enriched motif in the promoters of cluster 7 differentially expressed genes. For flowering time results, the bottom and top lines of the box indicate the first and third quantiles, and the bottom and top lines of the whisker denote minimum and maximum values. The bar inside the box is the median value (n=16). Asterisks denote significant differences from WT (***p<0.001, one-way ANOVA and Dunnett's multiple comparison test). (**B**) A Venn diagram showing the significantly down-regulated genes from bulk RNA-seq of two independent *pSUC2:NIGT1.2* and *pSUC2:NIGT1.4* lines compared with wild-type plants. (**C**) The effect of *NIGT1.2*, *NIGT1.4*, and *GFP* on wild-type and mutated *FT* promoter activity in tobacco transient assay. The 1 kb of *FT* promoter (*pFT*) and the same *FT* promoter with all NIGT1-binding sites mutated (*pFTm*) control the expression of firefly luciferase (*LUC*) gene. The LUC activity was normalized with Renilla luciferase (RLUC) activity controlled by *35* S promoter. The results are means ± SEM with each dot representing biological replicates (n=4 or 5). (**D**) Relative gene expression levels of *FT* and *BFT* in 14 day-old wild-type (WT) and *nigtQ* seedlings at ZT4. Plants were grown with

*Figure 5 continued on next page*

*Figure 5 continued*

high nitrogen (+N) and without (–N) in LD +FR. The results are means ± SEM with each dot representing biological replicates (n=6). Asterisks denote significant differences from WT (*p<0.05, **p<0.01, ***p<0.001, t-test). n.s. indicates not significant.

The online version of this article includes the following source data and figure supplement(s) for figure 5:

**Source data 1.** Enriched DNA motifs and potential transcription factors bind to the motifs in the promoters of 268 genes highly expressed in cluster 7.

**Source data 2.** Transcription factors that bind to four tandem repeats of CORE-containing sequences which also contain NIGT1-binding motifs of the FT promoter in yeast one-hybrid analysis.

The TF library containing 1957 genes was used to screen the binding TFs to the sequences.

**Source data 3.** A list of differentially expressed genes in *pSUC2:NIGT1.2* #1 compared with wild-type plants (n=3).

**Source data 4.** A list of differentially expressed genes in *pSUC2:NIGT1.2* #3 compared with wild-type plants (n=3).

**Source data 5.** A list of differentially expressed genes in *pSUC2:NIGT1.4* #3 compared with wild-type plants (n=3).

**Source data 6.** A list of differentially expressed genes in *pSUC2:NIGT1.4* #8 compared with wild-type plants (n=3).

**Figure supplement 1.** The 400 bp upstream regions of *FT* promoter sequences (**A**) and the NIGT-binding site mutated *FT* promoter sequences (**B**).

**Figure supplement 2.** Relative expression levels of *NIGT1.2*, *NiGT1.4,* and cluster 7-enriched genes (*FT*, *BFT*, *FLP1*, *PARCL*, *ROXY10*, AT2G26695, and AT1G67865) in 14-day-old wild-type (WT), *pSUC2:NIGT1.2*, and *NIGT1.4* plants at ZT4.

**Figure supplement 3.** Expression of *NIGT1.2* and *NIGT1.4* genes and their roles in flowering time.

We performed a yeast one-hybrid (Y1H) screen of 1957 TFs against four tandem repeats of a short *FT* promoter sequence containing the *S1* and *S2* domains and the two NIGT1 binding sites (*Bonaldi et al., 2017*; *Li et al., 2019*). We found that NIGT1.1/HHO3, NIGT1.2/HHO2, NIGT1.3/HHO1, NIGT1.4/HRS1, and HHO5 specifically bind to this sequence within the *FT* regulatory region (*Figure 5—source data 2*). Next, we overexpressed five different *NIGT1/HHO* genes using the *SUC2* promoter. At the $T_1$ generation, we observed the late flowering phenotype in the *pSUC2:NIGT1.2* and *NIGT1.4* lines, while it appeared less clear in the *pSUC2:NIGT1.1*, *NIGT1.3*, and *HHO5* (data not shown). Therefore, we generated $T_3$ lines of *pSUC2:NIGT1.2* and *pSUC2:NIGT1.4* and measured their flowering time. Under LD +FR conditions, these transgenic plants flowered significantly later than wild-type plants (*Figure 5A*). To test which genes were altered by the overexpression of *NIGT1.2* and *NIGT1.4*, we conducted RNA-seq analysis of the respective transgenic plants (*Figure 5B*, *Figure 5—source data 3–6*). We found that genes specifically expressed in cluster 7, such as *FT*, *BFT*, and *SAQR*, were downregulated in both the *pSUC2:NIGT1.2* and *pSUC2:NIGT1.4* lines. *FLP1* expression was decreased only in *pSUC2:NIGT1.2* lines, which might explain why *pSUC2:NIGT1.2* plants exhibited a more severe late flowering phenotype than *pSUC2:NIGT1.4* plants. Similar results were obtained with qRT-PCR analysis (*Figure 5—figure supplement 2*). To test if NIGT1.2 and NIGT1.4 directly regulate the *FT* promoter activity, we conducted the tobacco transient promoter luciferase (LUC) assay using the *FT* promoter (*pFT*) and the one with the mutations in three potential NIGT1 binding sites (*pFTm*) (*Figure 5—figure supplement 1B*). Our data showed that co-expression of NIGT1.2 and NIGT1.4 significantly decreased the promoter activity of *pFT* compared with that of *pFTm*, while GFP did not change the *pFT* activity compared with *pFTm* (*Figure 5C*), suggesting that NIGT1s directly repress *FT* expression level. Consistent with our snRNA-seq data (*Figure 5—figure supplement 3A and B*), previous studies reported that *NIGT1.2* is expressed broadly in shoots while *NIGT1.4* is expressed only in roots (*Kiba et al., 2018*), suggesting that *NIGT1.2* is the major player in *FT* regulation. Interestingly, neither *NIGT1.2* nor *NIGT1.4* expressed highly in cluster 7. We speculate that they have a role to prevent the misexpression of *FT* and perhaps other cluster 7 genes in non-cluster 7 cells.

Ample nitrogen prolongs the vegetative growth stage and delays flowering ; however, the detailed mechanisms of this delay remain largely unknown (*Sanagi et al., 2021*). Since *NIGT1* genes are induced under high nitrogen conditions, we tested the quadruple mutant of the *NIGT1* genes (*nigtQ* mutant) for *FT* and *BFT* expression with (+N) and without high nitrogen (–N) (*Kiba et al., 2018*). The *nigtQ* plants showed enhanced *FT* and *BFT* expression compared to the wild-type plants under +N but not –N conditions, suggesting that the NIGT1 TFs act as novel nitrate-dependent regulators of flowering (*Figure 5D*, *Figure 5—figure supplement 3C*). To test if *nigtQ* mutations affect flowering timing under different nitrogen conditions, we transplanted 9-day-old WT and *nigtQ* seedlings grown on Murashige and Skoog-based media with 20 mM $NH_4NO_3$ to new media containing either 20 mM or 2 mM $NH_4NO_3$. The *nigtQ* plants bolted earlier in days than WT when they grew with 20 mM $NH_4NO_3$,

whereas the flowering time of WT and *nigtQ* was almost identical under 2 mM, suggesting that *NIGT1* genes delay flowering under ample nitrogen conditions (*Figure 5—figure supplement 3D and E*). However, there was no difference between WT and *nigtQ* in leaf number at bolting (*Figure 5—figure supplement 3F*). Therefore, it appears that the *nigtQ* mutation enhanced overall plant growth speed rather than developmental timing of flowering. We also have counted leaf numbers of the *nigtQ* at bolting on nitrogen-rich soil. The mutant generated slightly more leaves than WT when they flowered (*Figure 5—figure supplement 3G*). These results suggest that the NIGT-derived fine-tuning of *FT* regulation is conditional on higher nitrogen conditions.

## Discussion

Plants utilize seasonal information to determine the onset of flowering. One of the key mechanisms of seasonal flowering is the transcriptional regulation of the florigen *FT* in leaves. Our previous study revealed that *FT*-expressing cells also produce FLP1, another small protein that may systemically promote both flowering and elongation of leaves, hypocotyls, and inflorescence stems (*Takagi et al., 2025*). Therefore, a more detailed characterization of *FT*-expressing cells might identify additional components that are essential for the integration of the many environmental cues in natural environments into the precise timing of flowering and related developmental changes. Here, we applied bulk nuclei and snRNA-seq and transgenic analysis to examine *FT*-expressing cells at high resolution.

### Differences in the spatial expression patterns of *FT* between cotyledons and true leaves are likely driven by negative regulators

In young *Arabidopsis* plants grown in LD, true leaves show *FT* expression in the distal and marginal parts of the leaf vasculature, while cotyledons express *FT* across entire veins (*Ito et al., 2012*; *Takada and Goto, 2003*). We used bulk RNA-seq to examine the *FT*-expressing cells in cotyledons versus those in true leaves, using FANS-enriched nuclei. Tissue-specific differences (cotyledons vs. true leaves) in gene expression were more significant than those observed for the enriched cell populations, an unexpected result as cotyledons and true leaves are not typically treated as separate organs.

The true leaves, but not the cotyledons, of the *pSUC2:NTF* and *pFT:NTF* lines showed differences in the expression of *FT* negative regulators. This result suggests that *FT* expression is strongly repressed in most of the true leaf companion cells, contributing to the spatial expression pattern of *FT* observed in true leaves. In this study, we focused on the *FT* expression peak in the morning (ZT4) under LD +FR conditions; however, in the future, it would be interesting to investigate the expression patterns of *FT* regulators at the *FT* evening peak (ZT16), as our results suggest that there are two different mechanisms regulating each *FT* peak in natural long days (*Song et al., 2018*; *Lee et al., 2023*).

### A subpopulation of phloem companion cells highly expresses *FT* and other genes encoding small proteins

Our snRNA-seq identified a cluster with nuclei derived from cells with high *FT* expression that controls flowering time. The cluster 7 nuclei are derived from cells that appear to be metabolically active and produce ATP, which may be used to upload sugars, solutes, and small proteins, including FT, into the phloem sieve elements. Based on subclustering analysis, *FT* expression levels were positively correlated with companion cell marker gene expression and negatively correlated with mesophyll cell marker gene expression. The observed shift from mesophyll to companion cell identity may represent the trajectory of companion cell development (*Torii et al., 2022*).

In clusters 4 and 5, we found nuclei isolated from phloem cells with high expression of aquaporin and JA-related genes, respectively. These cell populations were not identified in a previous protoplast-based scRNA-seq analysis targeting phloem (*Kim et al., 2021*). We speculate that these cells are highly embedded in the tissue and therefore practically difficult to isolate by protoplasting. Previous studies show that JA biosynthetic genes such as *LOX3/4* and *OPR3* are highly expressed in phloem including companion cells (*Chauvin et al., 2016*; *Li et al., 2013*), consistent with our results.

Aquaporin genes such as *PIP2;1* and *PIP2;6* are thought to be highly expressed in leaf xylem parenchyma and bundle sheath tissues (*Prado et al., 2013*). However, cluster 6, which contains nuclei isolated from bundle sheath and phloem parenchyma cells, showed fewer expressed aquaporin genes than the cluster 4 nuclei, which also expressed *SUC2*. This result might indicate that the cells active in

water transport exist not only in bundle sheath or xylem parenchyma cells but also in a subpopulation of phloem cells. In summary, *SUC2/FT*-expressing vasculature cells consist of cell subpopulations with different functions within true leaves.

A drawback of our snRNA-seq analysis was shallow reads per nucleus. It appears mainly due to the low abundance of mRNA in FANS-isolated nuclei from 2-week-old leaves. Based on our calculation, the average mRNA level per nucleus in our isolated samples is approximately 0.2 pg (3000 pg mRNA from 15,000 sorted nuclei). Future technological advances are needed to improve the data quality.

## BFT may fine-tune the balance between flowering and growth as a systemic anti-florigen

The *FT*-expressing nuclei of cluster 7 preferentially expressed genes encoding small proteins including *FLP1*, *BFT,* and *SAQR* (*Yoo et al., 2010*; *Ryu et al., 2014*; *Jones et al., 2016*). Although a previous study reported that *BFT* driven by the *SUC2* promoter does not alter flowering time (*Ryu et al., 2014*), our result shows that overexpression of this gene delays flowering under LD +FR conditions (*Figure 4C*). This discrepancy is likely due to our use of growth conditions with the adjusted R/FR ratio that mimics natural light conditions, as the majority of our transgenic lines showed a similar late flowering phenotype.

Why do *Arabidopsis* plants produce florigen FT and anti-florigen BFT in the same cells? There is precedent for the co-expression of florigen and anti-florigen in leaves in other plants. In Chrysanthemums, a weak florigen *CsFTL1* and a strong anti-florigen *CsAFT* are both expressed in leaves in long days. Short-day conditions repress *CsAFT* but induce the expression of a strong florigen *CsFTL3* to initiate flowering (*Higuchi, 2018*; *Higuchi et al., 2013*). The balance in florigen and anti-florigen expression is also important for wild tomatoes to control flowering time, and this regulation was lost in domesticated tomatoes, resulting in day-neutral flowering behaviors (*Soyk et al., 2017*). In *Arabidopsis*, the FT-homolog floral repressor TERMINAL FLOWER 1 (TFL1) is expressed at the shoot apical meristem and competes with FT for physical interaction with FD (*Zhu et al., 2020*). In addition to preventing precocious flowering, the presence of TFL1 is crucial for the balance between flowering initiation and stem growth because FT terminates inflorescence stem growth (*McKim, 2020*). Like FT and TFL1, BFT directly binds to FD (*Ryu et al., 2014*), suggesting that BFT's mode of action is similar to that of TFL1. However, unlike for TFL1, loss of BFT does not affect flowering time under standard growth conditions (*Ryu et al., 2014*; *Yoo et al., 2010*). The lack of a visible flowering phenotype in the *bft* knockout mutant might be due to the substantially lower expression of *BFT* compared to *FT* under standard growth conditions (*Ryu et al., 2014*). Consistent with this interpretation, the *bft* knockout mutant is significantly delayed in flowering under high salinity conditions where *BFT* gene expression is strongly induced (*Ryu et al., 2014*). Thus, BFT appears to act as an anti-florigen under specific growth conditions, possibly including LD +FR conditions. Given the complexity of natural environments, the existence of multiple specialized anti-florigens might facilitate signal integration and reduce noise in the onset of flowering and stem growth.

## NIGT1 TFs contribute to the nutrient dependency of flowering time

To identify potential novel transcriptional regulators of *FT*, we identified enriched motifs in the promoters of the 268 genes that were differentially expressed in cluster 7. This analysis indicated that NIGT1 TFs may affect the expression of *FT* and other genes co-expressed with *FT*. Using Y1H screening, we found that all NIGT1s and HHO5 can bind to the enhancer sequences derived from the *FT* promoter. NIGT1s are involved in nitrogen absorption as negative regulators (*Kiba et al., 2018*; *Maeda et al., 2018*). It is well-known that nitrogen availability affects the onset of flowering; however, most of the mechanistic underpinnings remain unknown. A recent study showed that ample nitrogen availability causes phosphorylation of FLOWERING BHLH 4 (FBH4), a direct positive regulator of *CO*, which results in attenuation of FBH4 transcriptional activity (*Sanagi et al., 2021*). This mechanism acts upstream of *FT*, whereas the NIGT1 TFs likely act as direct repressors of *FT*. Multiple nitrogen-dependent mechanisms acting on *FT* regulation might ensure the proper balance between nutrient availability and resource allocation toward developmental transitions.

# Materials and methods

**Key resources table**

| Reagent type (species) or resource | Designation | Source or reference | Identifiers | Additional information |
|---|---|---|---|---|
| Gene (*Arabidopsis thaliana*) | *FT* | TAIR | AT1G65480 | |
| Gene (*Arabidopsis thaliana*) | *SUC2* | TAIR | AT1G22710 | |
| Gene (*Arabidopsis thaliana*) | *CAB2* | TAIR | AT1G29920 | |
| gene (*Arabidopsis thaliana*) | NIGT1.2 | TAIR | AT1G68670 | |
| Gene (*Arabidopsis thaliana*) | *NIGT1.4* | TAIR | AT1G13300 | |
| Gene (*Arabidopsis thaliana*) | *BFT* | TAIR | AT5G62040 | |
| Gene (*Arabidopsis thaliana*) | *FLP1* | TAIR | AT4G31380 | |
| Gene (*Arabidopsis thaliana*) | *PIP2;6* | TAIR | AT2G39010 | |
| Gene (*Arabidopsis thaliana*) | *ROXY10* | TAIR | AT5G18600 | |
| Strain, strain background (*Escherichia coli*) | TOP10 | Invitrogen | C404010 | |
| Strain, strain background (*Agrobacterium tumefaciens*) | GV3101 | Imaizumi lab | | |
| Genetic reagent (*Arabidopsis thaliana*) | nigtQ | **Kiba et al., 2018** | | |
| Genetic reagent (*Arabidopsis thaliana*) | ft-101 | **Takada and Goto, 2003** | | |
| Genetic reagent (*Arabidopsis thaliana*) | pFT:NTF | **Takagi et al., 2025** | | |
| Genetic reagent (*Arabidopsis thaliana*) | *pSUC2:NTF, pCAB2:NTF, and p35S:NTF* | This study | | See molecular cloning and plant materials in Materials and methods |
| Genetic reagent (*Arabidopsis thaliana*) | *pBFT:H2B-tdTomato, pAT1G24575:H2B-tdTomato, pAT1G67865:H2B-tdTomato, pROXY10:H2B-tdTomato, pAT2G26695:H2B-tdTomato, pPARCL:H2B-tdTomato, pROXY7:H2B-tdTomato, and pCYSTM12:H2B-tdTomato all in pFT:NTF background* | This study | | See molecular cloning and plant materials in Materials and methods |
| Genetic reagent (*Arabidopsis thaliana*) | *pSUC2:GFP, pSUC2:BFT, pSUC2:AT1G24575, pSUC2:AT1G67865, pSUC2:ROXY10, pSUC2:AT2G26695, pSUC2:SAQR, pSUC2:RL4, pSUC2:AT1G54575, pSUC2:NIGT1,2, and pSUC2:NIGT1.4 all in pFT:GUS background* | This study | | See molecular cloning and plant materials in Materials and methods |
| Genetic reagent (*Arabidopsis thaliana*) | *pPIP2;6:FLAG-GFP-RPL18* | This study | | See molecular cloning and plant materials in Materials and methods |
| Genetic reagent (*Arabidopsis thaliana*) | *pROXY10:amiR-ft, pSUC2:amiR-ft, pPIP2;6:amiR-ft, and pGC1:amiR-ft* | This study | | See molecular cloning and plant materials in Materials and methods |
| Recombinant DNA reagent | *pFT:LUC and pFTm:LUC* | This study | | See tobacco transient promoter LUC assay in Materials and methods |
| Recombinant DNA reagent | *p35S:NIGT1.2, p35S:NIGT1.4, and p35S:GFP* | This study | | See tobacco transient promoter LUC assay in Materials and methods |
| Recombinant DNA reagent | pENTR-/D-TOPO | Invitrogen | Cat# K230020SP | |

| Reagent type (species) or resource | Designation | Source or reference | Identifiers | Additional information |
|---|---|---|---|---|
| Commercial assay or kit | Dual Luciferase Reporter Assay System | Promega | Cat# E1910 | |
| Commercial assay or kit | PrimeScript RT reagent Kit | Takara Bio | Cat# RR037A | |
| Commercial assay or kit | SMART-seq v4 3' DE Kit | Takara Bio | Cat# 635040 | |
| Commercial assay or kit | KAPA SYBR FAST qPCR Master Mix (2 x) kit | Roche | Cat# KK4602 | |
| Commercial assay or kit | Chromium Next GEM Single Cell 3' GEM, Library & Gel Bead Kit v3.1 | 10X Genomics | Cat# PN-1000128 | |
| Commercial assay or kit | Chromium Next GEM Chip G Single Cell Kit | 10X Genomics | Cat# PN-1000127 | |
| Chemical compound, drug | Cellulase 'onozuka' R-10 | Yakult Pharmaceutical Industry | Cat# 636–01441 | |
| Chemical compound, drug | Macerozyme R-10 | Yakult Pharmaceutical Industry | Cat# 635–02631 | |
| Chemical compound, drug | Glycogen | Thermo Fisher Scientific | Cat# R0551 | 20 mg/mL |
| Software, algorithm | Prism 10 | GraphPad | RRID: SCR_002798 | |
| Software, algorithm | Cellranger v3.0.1 | 10X Genomics | RRID: SCR_023221 | |
| software, algorithm | Seurat | *Hao et al., 2021* *Satija et al., 2015* | RRID: SCR_007322 | |
| Software, algorithm | Eukaryotic Promoter Database | *Dreos et al., 2015*, *Meylan et al., 2020* | RRID: SCR_002132 | |
| Software, algorithm | Simple Enrichment Analysis | *Bailey and Grant, 2021* | RRID: SCR_001783 | |
| Software, algorithm | STAR | *Dobin et al., 2013* | RRID: SCR_004463 | |
| Other | CellTrics 30 µm | Sysmex | Cat# BV264870 | See nuclei isolation by FANS in Materials and methods. |
| Other | SUPERase-In RNase Inhibitor | Thermo Fisher Scientific | Cat# AM2694 | See nuclei isolation by FANS in Materials and methods. |

## Molecular cloning, plant materials, and growth

All *Arabidopsis thaliana* transgenic plants and mutants are Col-0 backgrounds. *ACT2:BirA* plant (*Deal and Henikoff, 2011*) was used as the *Arabidopsis* genetic background to generate *NTF*-expressing lines. The *NTF* cDNA was controlled by the *35* S promoter or the tissue-specific promoters: *FT*, *SUC2*, and *CAB2*. The *NTF* sequences were amplified using primers (5'-CACCATGGATCATTCAGCGAAAAC CACACAG-3' and 5'- TCAAGATCCACCAGTATCCTCATGC-3') and cloned into the pENTR/D-TOPO vector (Invitrogen). The *SUC2* promoter region (2302 bp) was amplified by primers (5'-GGTGCATA ATGATGGAACAAAGCAC-3' and 5'-ATTTGACAAACCAAGAAAGTAAG-3') and cloned into the pENTR 5'-TOPO (Invitrogen, Waltham, MA). The *CAB2* promoter region (324 bp) was amplified by the primers (5'-CACCATATTAATGTTTCGATCATCAGAATC-3' and 5'- TTCGATAGTGTTGGATTATATAGG G-3') and cloned into the pENTR 5'-TOPO (*Maxwell et al., 2003*). The *SUC2* and *CAB2* promoter sequences were fused with *NTF* in the binary GATEWAY vector R4pGWB501 through LR clonase reaction (*Nakagawa et al., 2008*). The *SUC2:NTF* and *CAB2:NTF* constructs were transformed into wild-type Col-0. The *pFT:NTF* line was described in the previous study (*Takagi et al., 2025*).

The *H2B-tdTomato* constructs containing promoter regions of cluster 7 highly expressed genes were made by swapping the heat-shock promoter (*pHS*) of pPZP211 HS:H2B-tdTomato (*Takagi et al., 2025*). Promoter sequences were amplified by the forward primer containing SbfI site and reverse primer containing SalI site and inserted into these restriction enzyme sites. Following primer

sequences were used to amplify 2396 bp upstream of *BFT*, 5'- <u>CCTGCAGG</u>GACAGAGTAAATTCAA CCACAGCAGGT-3' and 5'- <u>GTCGAC</u>TTTTCTTTGCTCCAATGTGTTTGCGTTTG-3'; 2521 bp upstream of AT1G24575, 5'- <u>CCTGCAGG</u>CTCTCAGATCACCGTAAGGGCATAATTATATTTAGGTTCAC-3' and 5'- <u>GTCGAC</u>GTGATGAGATTTGTGACTGGAGGAGTTTCCAAGTACCATTCTT-3'; 2488 bp upstream of AT1G67865, 5'- <u>CCTGCAGG</u>ACTTCACATTCTTGGATTCCGTTTGTAATAACTAATGTTTT-3' and 5'-<u>GTCGAC</u>CCCTCCGGCAACCCCAATAATAAGCTTATCAAGCATTTTTCTT-3'; 1939 bp upstream of *ROXY10*, 5'-<u>CCTGCAGG</u>GCAATGGACCGTACGTCTAGGTCACGCATCTTATCCGACAT-3' and 5'- <u>GTCGAC</u>CACCGGTCTCTCCATCACCATCTTCGTTATCATATCCATTGCT-3'; 2226 bp upstream of AT2G26695, 5'-<u>CCTGCAGG</u>AATGTAATGTATAATGTGTTCATAAACAGCACCAACTACCC-3' and 5'- <u>GTCGAC</u>TGCGTGCTGACACGCACCACATAGCCAATCTCCTCCGGTCCAG-3'; 1911 bp upstream of *PARCL* (AT1G64370), 5'-<u>CCTGCAGG</u>GCCCATCTAATTCCCATTTTAGATGCATGAGTTCAACGCT A-3' and 5'- <u>GTCGAC</u>GCCTTGAGCCACCTCGTAGTAGTCTTTCTCACGGTTTTCGTAG-3'; 2719 bp upstream of *ROXY7* (AT2G30540), 5'-<u>CCTGCAGG</u>ATCACCGGTAAGTGACAAGAGAATTGA-3' and 5'-<u>GTCGAC</u>GGTTTCTTGAAGGAGGTCTCGATCAATCT-3'; 2574 bp upstream of *CYSTM12* (AT5G04080),5'- <u>CCTGCAGG</u>GAGAATTTGAAGGAGGCTTTGCGTTTTATCTGCTCATCTAA-3' and 5'- <u>GTCGAC</u>TTGTGGAGGATTTTGATCTCTCATGTCCTGCATCTTCTCAAAA-3'. These constructs were transformed into *pFT:NTF* plants.

To overexpress cluster 7 highly expressed genes, we amplified coding sequences using the following primer sets. *BFT*, 5'-CACCATGTCAAGAGAAATAGAGCCAC-3' and 5'-AGTTAATAAGAAGGACGTCG TCG-3'; AT1G24575, 5'- CACCATGGTACTTGGAAACTCCTCCAGTCAC-3' and 5'- GACTAGGC GCTCTTAGTCATCCAC-3'; AT1G67865, 5'- CACCATGCTTGATAAGCTTATTATTGGGGTTGCC-3' and 5'-CTTTACTCCCTGTCTTTCTGGCG-3'; *ROXY10*, 5'- CACCATGGATATGATAACGAAGATGGTG ATGGAG-3' and 5'- AGTCAAACCCACAATGCACCAG-3'; AT2G26695, 5'- CACCATGAGCTGGACC GGAGGAG-3' and 5'-ATTTAGACGCCACCATAATCTCTTG-3', *SAQR* (AT1G64360), 5'-CACCATGT CGTTTAGAAAAGTAGAGAAGAAACC-3' and 5'- GATTAGTAATTAGGGAAGTGTTTGCGGC-3'; *RL4* (AT2G18328), 5'-CACCATGGCTTCTAGTTCAATGAGCACC-3' and 5'- CTTCAATTAGTGTTACGGTA CCTAGG-3'; AT1G54575, 5'- CACCATGGTGGATCATCATCTCAAAGC-3' and 5'- AATTAATTATTC TTTTGTGGCTTGG-3'. Amplified cDNA was cloned into pENTR/D-TOPO, followed by GATEWAY cloning using pH7SUC2, a binary vector carrying 0.9 kb *SUC2* promoter (*Takagi et al., 2025*). These vectors were transformed into wild-type plants possessing *pFT:GUS* reporter gene (*Takada and Goto, 2003*).

To generate the *pPIP2;6:FLAG-GFP-RPL18* line, we first synthesized the GATEWAY destination vector with the *PIP2;6* promoter (pH7PIP2;6). *PIP2;6* upstream sequences (3110 bp) were cloned as two overlapped fragments for ease in amplification. The *PIP2;6* fragment 1 (−3110—1566 bp positions from ATG) was amplified by the forward primer containing HindIII site (5'- <u>AAGCTT</u>CACCATAT GACCACCACCATCATCAACATCATCCATCATCATCTTCACCC-3') and the reverse primer (5'- GCGA AGCATTTGGCGGATTATAGAGTTCTACAAGACTACAACAGATGATGCATCA-3'), and the fragment 2 (−1706—1) was amplified by the forward primer (5'-TTATGAACGGTCCCATCTCTAGGAAAAATGAG TAATATAATTCATGAAGCAATTCATT-3') and the reverse primer containing SpeI site (5'- <u>ACTAGT</u>TCTT TCAGACTTAGCCTTCACGGACTCAAAAAAGAAAGAGAGAGAGAGAAGAGAG-3'). Subsequently, fragment 1 and fragment 2 were digested by HindIII and BspHI (BspHI site exists in overlapped sequences of both fragments), and BspHI and SpeI, respectively, and the destination vector backbone pH7WG2 was digested with HindIII and SpeI to remove its *35* S promoter (*Karimi et al., 2002*). Next, two digested fragments of *PIP2;6* were ligated into pH7WG2 without *35* S promoter (the resulting product was pH7PIP2;6). *FLAG-GFP-RPL18* was amplified using *pCER5:FLAG-GFP-RPL18* plasmid (*Mustroph et al., 2009*) as a template and primers (5'- CACCTATTTTTACAACAATTACCAACAAC-3' and 5'- TTAAACCTTGAATCCACGACTC-3'), subsequently cloned into the pENTR/D-TOPO. Then *FLAG-GFP-RPL18* in pENTR/D-TOPO was integrated into pH7PIP2;6 binary vector.

To repress *FT* expression in specific cell types, we generate *pROXY10:amiR-ft, pSUC2:amiR-ft, pPIP2;6:amiR-ft, and pGC1:amiR-ft* lines. First, we synthesized the GATEWAY destination vectors with the *ROXY10* and *GC1* promoters (pH7ROXY10 and pH7GC1). Promoter sequences of *ROXY10* and *GC1* were amplified by the forward primer containing HindIII site and the reverse primer containing SpeI site, and inserted into these restriction enzyme sites of pH7WG2 to replace its *35* S promoter. Following primer sequences were used to amplify 1900 bp upstream of *ROXY10*, 5'- <u>AAGCTT</u>GCAA TGGACCGTACGTCTAGGTCACGC –3' and 5'- <u>ACTAGT</u>TGCTATTTTATATGGTATATGATCCAACA –3';

and 1,163 bp upstream of *GC1*, 5'- <u>AAGCTT</u>ATGGTTGCAACAGAGAGGATGAATTTATA and 5'- <u>ACTA</u><u>GT</u>ATTTCTTGAGTAGTGATTTTGAAGTAGTGTGTGAAAATAGTAC-3'. The *amiR-ft* sequences were synthesized through the overlap extension PCR using pRS300 as a template (*Schwab et al., 2006*). Following primer sets were used to amplify fragment #1, 5'- CACCCTGCAAGGCGATTAAGTTGG GTAAC-3' and 5'- GAACCAAAGAATAGAAGTTCTAGTCTACATATATATTCCT-3'; fragment #2, 5'- GACTGGAACTTCTATACTTTGGATCAAAGAGAATCAATGA-3' and 5'- GACTAGAACTTCTATTCTTT GGTTCACAGGTCGTGATATG-3'; fragment #3, 5'- GATCCAAAGTATAGAAGTTCCAGTCTCTCTTT TGTATTCC-3' and 5'- GCGGATAACAATTTCACACAGGAAACAG-3'. Next, fragments #1–3 were mixed to run overlap extension PCR. The PCR product containing *amiR-ft* was cloned into pENTR/D--TOPO. Finally, *amiR-ft* in pENTR/D-TOPO was integrated into pH7ROXY10, pH7SUC2, pH7PIP2;6, and pH7GC1 binary vectors.

For gene expression measurements and confocal microscope imaging, surface-sterilized seeds were sawn on the 1 x Linsmaier and Skoog (LS) media plates without sucrose containing 0.8% (w/v) agar and stratified for more than 2 days at 4 °C before being transferred to the incubator. Plants were grown under LD +FR conditions (100 μmol photons m$^{-2}$ s$^{-1}$, red/far-red ratio = 1.0) for 2 weeks as described previously (*Takagi et al., 2025*). For gene expression analysis under –N conditions, plants were grown on normal 1 x Murashige and Skoog (MS) medium plates with full nitrogen (+N) for 10 days and transplanted to +N and –N medium plates and grown for 4 additional days. The ion equilibrium of the medium between +N and –N was ensured by replacing $KNO_3$ (18.79 mM) and $NH_4NO_3$ (20.61 mM) with KCl (18.79 mM) and NaCl (20.61 mM). For flowering time measurements using T$_1$ transgenic lines (*Figure 4C*), T$_1$ plants were grown on hygromycin selection plates for 10 days and transferred to Sunshine Mix 4 soil (Sun Gro Horticulture, Agawam, MA). Soil was supplemented with a slow-release fertilizer (Osmocote 14-14-14, Scotts Miracle-Gro, Marysville, OH) and a pesticide (Bonide, Systemic Granules, Oriskany, NY) and filled in standard flats with inserts (STF-1020-OPEN and STI-0804, T.O. Plastics, Clearwater, MN). For the same experiment using *pSUC2:NIGT1* T$_3$ plants (*Figure 5A*), surface-sterilized seeds were directly sawn on soil and kept in 4 °C for at least 2 days. Plants on soil were kept grown under LD +FR conditions as described previously (*Takagi et al., 2025*). Tissue clearing and confocal imaging were conducted as previously described (*Takagi et al., 2025*).

## Nuclei isolation by FANS

Nuclei labeled with NTF protein were isolated with the modified method described by *Galbraith, 2014*. Cotyledons and true leaves were detached from 2-week-old plants grown under LD +FR conditions at ZT4 (approximately 100–200 seedlings were used at each sample), and placed on a mixture of 2 mL filter sterilized chopping buffer (20 mM MOPS, 30 mM sodium citrate and 45 mM $MgCl_2$) and 20 μL SUPERase-In RNase inhibitor (20 U /μL) (Thermo Fisher Scientific, Waltham, MA; *Figure 1—figure supplement 1B and C*). We did not use fixing reagents such as formaldehyde in the chopping buffer because it caused clumps of nuclei. Leaves were chopped with the razor blade in a 4 °C room until finely crushed in the buffer solution, followed by the addition of 3 mL more chopping buffer and filtration with a 30 μm CellTrics filter (Sysmex, Kobe, Japan). It usually takes approximately 30 min from start chopping leaves to applying the samples for sorting. The filtrated sample solution was directly applied to the sorting with SH800S Cell Sorter equipped with a 100 μm flow tip (SONY Biotechnology, San Jose, CA). As a sheath solution, we used 0.9% NaCl but not PBS to prevent calcium precipitation by phosphate contained in PBS, which severely decreases the integrity of nuclei. DEPC-treated water was used for all buffers and solutions in this experiment to prevent RNA degradation. Nuclei were excited by 488 nm collinear laser, and FITC and mCherry channels were used to detect GFP signal and autofluorescence, respectively. The area of GFP-positive nuclei was determined using the parental line *pACT2:BirA* (*Figure 1—figure supplement 2*).

## SMART-seq2 library preparation and analysis

For bulked nuclei RNA-seq, approximately 15,000 nuclei were collected from cotyledons and true leaves of *pSUC2:NTF*, *pCAB2:NTF*, and *p35S:NTF* lines, while 10,000 (cotyledons) and 3000 (true leaves) nuclei were collected from *pFT:NTF* due to fewer population of GFP-positive nuclei. Sorted nuclei were directly collected into Buffer RTL (QIAGEN, Hilden, Germany), and RNA was extracted according to the manufacturer protocol of RNeasy Micro Kit (QIAGEN). Three independent biological replicates were produced for cotyledon and true leaf of all transgenic lines.

After RNA integrity was confirmed using High Sensitivity RNA Screen Tape (Agilent Technologies, Santa Clara, CA), SMART-seq2 libraries were generated according to the manufacturer's protocol (Takara Bio USA, San Jose, CA). Sequencing was performed on the Illumina NovaSeq 6000 platform through a private sequencing service (Novogene, Beijing, China). Since reverse reads were undesirably trimmed due to the presence of in-line indexes in SMART-seq libraries, we used only forward reads of paired-end reads. Approximately 10.8–34.6 million forward reads (average, 20.2 million reads) were produced from each sample. By STAR software (version 2.7.6 a; *Dobin et al., 2013*), reads were mapped to *Arabidopsis* genome sequences from The *Arabidopsis* Information Resource (TAIR, version 10; http://www.arabidopsis.org/). DEGs were selected based on the adjusted p-value calculated using DEseq2 in the R environment.

## Single-nucleus RNA-seq library preparation

For snRNA-seq experiments, sorted nuclei from true leaves were collected in the mixture of 100 μL 1 x PBS and 10 μL SUPERase-In RNase inhibitor in a 15 mL Corning tube. Prior to nuclei collection, the corning tube was coated by rotating with 10 mL 1 x PBS containing 1% BSA at room temperature to prevent static electricity in the wall. After more than 10,000 nuclei were collected in the corning tube (sorting approximately for 1 hr), 10 μL of 1 mg/mL DAPI solution and 1/100 volume of 20 mg/mL glycogen (Thermo Fisher Scientific) were added and centrifuged in 1000x *g* for 15 min at 4 °C. Nuclei numbers were determined using a hemocytometer after resuspension with a small volume of 1 x PBS. The total number of nuclei varied depending on the plant line and trial but was no more than 11,000.

snRNA-seq was performed using the 10 x Single-cell RNA-Seq platform, the Chromium Single Cell Gene Expression Solution (10x Genomics, Pleasanton, CA). Two biological replicates of *SUC2:NTF* and one replicate of *pFT:NTF* were produced for a total of three samples.

## Estimating gene expression in individual nucleus

Sequencing of snRNA-seq reads was performed on the Illumina Nextseq 550 platform, followed by the mapping to the TAIR10 *Arabidopsis* genome using Cellranger version 3.0.1 software.

The Seurat R package (version 4.0.5; *Hao et al., 2021*; *Satija et al., 2015*) was used for the dimensional reduction of our SnRNA-seq data. To remove potential doublets and background, we first filtered out nuclei with less than 100 and more than 5500 detected genes, and more than 20,000 UMI reads. For UMAP data visualization and cell clustering, all biological replicates were combined, and twenty principal components were compressed using a resolution value of 0.5.

Significantly highly expressed genes in each cluster compared with the entire nuclei population were identified using FindMarkers in Seurat with default parameters.

To show the enrichments of specific sets of genes, average expression of ATP biosynthesis, JA responsive, aquaporin, bundle sheath, and phloem parenchyma marker genes were visualized using AddModuleScore in Seurat. For bundle sheath and phloem parenchyma markers, we leveraged the top 50 most highly expressed genes in these cell types in previous protoplast-based ScRNA-seq data (*Kim et al., 2021*).

## Protein amino acid length

Amino acid length of proteins encoded by genes highly expressed in clusters 4, 5, and 7 was obtained from the Bulk Data Retrieval tool in TAIR database (https://www.arabidopsis.org/tools/bulk/sequences/index.jsp).

## Promoter cis-enrichment analysis

To identify highly enriched *cis*-elements in cluster 7 highly expressed genes, promoter sequences (500 bp) of 268 genes highly expressed in cluster 7 and randomly selected 3000 genes by R coding were extracted using the Eukaryotic Promoter Database (*Dreos et al., 2015*; *Meylan et al., 2020*). Obtained promoter sequences were next submitted to Simple Enrichment Analysis (SEA) (*Bailey and Grant, 2021*) in MEME Suite server (version 5.4.1) to elucidate what *cis*-elements are enriched in cluster 7 highly expressed genes through a comparison with random 3000 genes.

## Preparation and gene expression analysis of leaf mesophyll protoplasts

Protoplasts were isolated from true leaves of 2-week-old plants grown under either LD or LD +FR conditions (*Song et al., 2018*) by the modification of the method for protoplast isolation from cotyledons (*Endo et al., 2016*). The first and second true leaves were used to generate protoplasts, and 4 true leaves were placed on an adhesive side of the tube labeling tape (Tough-Spots, 1/2" diameter for 1.5–2.0 mL tubes, Research Products International, Mt Prospect, IL). To remove the epidermis and expose mesophyll cells, leaves were sandwiched with two tube tapes and peel tapes apart. A total of eight epidermis-removed leaves per tube was incubated in 1 mL enzymatic solution [0.75% (w/v) cellulase 'onozuka' R-10, 0.25% (w/v) macerozyme R-10, 0.4 M mannitol, 8 mM $CaCl_2$ and 5 mM MES-KOH] supplemented with 10 μL RiboLock RNase Inhibitor (Thermo Fisher Scientific) for 50 min with gentle rotation. The enzymatic solution containing protoplasts was filtrated with CellTrics 30 μm (Sysmex, Kobe, Japan), followed by centrifugation at 100xg for 5 min at room temperature. The supernatant was removed and added RLT buffer of RNeasy Plant Mini kit (QIAGEN) for RNA isolation.

## Yeast one-hybrid screening

Four tandem repeats of S1/S2 elements of *FT* promoter that include a CO-responsive (CORE) element (*Adrian et al., 2010*; *Tiwari et al., 2010*) were generated through restriction enzyme-mediated ligation. Forward and reverse oligonucleotides containing S1/S2 element with HindIII (CORE-F1: 5'-<u>AGCT</u><u>T</u>ACTGTGTGATGTACGTAGAATCAGTTTTAGATTCTAGTA   CTGTGTGATGTACGTAGAATCAGTTTTA GATTCTA<u>G</u>-3'), EcoRI (CORE-R1: 5'-<u>AATTC</u>CTAGAATCTAAAACTGATTCTACGTACATCACACAG TACTAG<u>AATCTAAAACTGATTCTACGTACATCACACAGTA</u>-3'), SacI (CORE-F2: 5'-<u>C</u>ACTGTGTGATGT ACGTAGAATCAGTTTTAGATTCTAGTACTGTGTGATGTAC GTAGAATCAGTTTTAGATTCTAG<u>GGTAC</u>-3'), and KpnI (CORE-R2: 5'-<u>C</u>CTAGAATCTAAAACTGATTCTACGTACATCACACAGTACTAGAATCTA AAACTGATTCTACGTACATCACACAGT<u>GAGCT</u>-3') recognition sequences were chemically synthesized from Genewiz (South Plainfield, NJ). The oligonucleotides complementary to each other were denatured for 10 min at 95 °C, then annealed for 30 min at RT. The resultant product was inserted into pENTR/D-TOPO vector harboring MCS sites (HindIII, EcoRI, SacI, and KpnI) through restriction enzyme-mediated ligation. Four tandem S1/S2 elements in pENTR/D-TOPO were then inserted into pY1-gLUC59-GW vector (*Bonaldi et al., 2017*) using LR clonase II (Invitrogen). The resultant pY1-gLUC59-GW-2XCORE plasmids were used for Y1H screening analysis.

## qRT-PCR analysis using total RNA

Total RNA extraction, cDNA synthesis, and qRT-PCR were performed as previously described (*Takagi et al., 2023*). *ISOPENTENYL PYROPHOSPHATE / DIMETHYLALLYL PYROPHOSPHATE ISOMERASE* (*IPP2*) and *PROTEIN PHOSPHATASE 2 A SUBUNIT A3* (*PP2AA3*) were used as reference genes. For statistical tests, relative expression levels were $\log_2$-transformed to meet the requirements for homogeneity of variance. qPCR primers used in this study are listed in *Table 1*.

## RNA-seq analysis using total RNA

Two-week-old seedlings of *pSUC2:NIGT1.2* and *NIGT1.4* grown on 1xLS media plates under LD +FR conditions were harvested at ZT4. RNA extraction and RNA-seq library preparation were conducted as described previously (*Takagi et al., 2025*; *Suzuki et al., 2022*). As a reference, our previous gene expression data of wild-type plants grown at the exact same conditions was compared with *pSUC2:NIGT1* lines (*Takagi et al., 2025*).

## Tobacco transient promoter LUC assay

We amplified 1 kb upstream of *FT* (*pFT*) using the primer sets, pFT-FW (5'- CACCATAATATGGCCG CTTGTTTATA-3') and pFT-RE (5'- CTTTGATCTTGAACAAACAGGTGG-3'), and cloned it into pENTR/D-TOPO. To mutate 2 of 3 potential NIGT1-binding sites, we synthesized the Megaprimer #1 by PCR using the forward primer pFTmut1 (5'- CAATGTGTGATGTACGTATACTCAGTTTTAGAGTATAGTA CATCAATAGACAAGAAAAG-3') and pFT-RE. Next, to mutate the rest of the NIGT1-binding site, the Megaprimer #2 was synthesized using the forward primer pFTmut2 (5'-CTACCAAGTGGGAGAT ATAATTTGAATTAATTCCAGTGTATTAGTGTGGTG) and Megaprimer #1. Subsequently, we amplified 1 kb promoter *FT* with mutations in all 3 NIGT1-binding sites (*pFTm*), we amplified DNA using pFT-FW and Megaprimer #2 and cloned it into pENTR/D-TOPO. Finally, *pFT* and *pFTm* regions were cloned

**Table 1.** qRT-PCR primers used in this study.

| Annotation | AGI code | Forward (5'→3') | Reverse (5'→3') |
|---|---|---|---|
| PP2AA3 | AT1G13320 | GCGGTTGTGGAGAACATGATACG | GAACCAAACACAATTCGTTGCTG |
| IPP2 | AT3G02780 | GTATGAGTTGCTTCTCCAGCAAAG | GAGGATGGCTGCAACAAGTGT |
| FT | AT1G65480 | CTGGAACAACCTTTGGCAAT | TACACTGTTTGCCTGCCAAG |
| SUC2 | AT1G22710 | GTGGGAGGTGGACCATTCGACG | CCGGAGGCGGTGAAGGCAAC |
| AHA3 | AT5G57350 | GGCTCATGCACAAAGGACTTTACACG | GCGATCTCAGCTCGTCTCTTGGC |
| Sultr2.1 | AT5G10180 | GGTGTTGAGCTAGTGATCGTTAACCCG | CCCGTAACACAACTGGTCCTTTGA |
| RBCS1A | AT1G67090 | GGCCTCCGATTGGAAAGAAGAAG | GGTGTTGTGCGAATCCGATGATCCTA |
| LHCB2.1 | AT2G05100 | TTGGTGTATCCGGTGGTGGCC | GTCCGTACCAGATGCTTTGAGGAGTAGA |
| GC1 | AT1G22690 | TCGTCCAAGAATCAATTGTGGGC | GTGTTGCCGGAGGTTCCCGG |
| CER5 | AT1G51500 | AGGAATATCGCTCGAGATGG | TGTCTCCCGAATCCTTTGAG |
| ML1 | AT4G21750 | CTACTCACAGTTGCGTTTCAGATAC | CCTTCCGAAAACATCGATTAGGCTC |
| NIGT1.2 | AT1G68670 | AAACCAAAAGCGGTGCGTT | ACTAGCTACTTTCACCGCCG |
| NIGT1.4 | AT1G13300 | CTAACAACGGAAACTCTCAAACG | CGGTAGTCTTGCCCGTAGAGTA |
| BFT | AT5G62040 | CCCGAGTCCTAGTAATCCTTATATGCGTGAAT | TATCTGTGTATTCCCGCCACCGGTTT |
| FLP1 | AT4G31380 | TCCAGTTCCACAAGAGAACTTCG | TCACGGACATGGAAGACGTTAGG |
| PARCL | AT1G64370 | AGAGTCAAGGACACATGGGAAGCTCTT | TCACCATTGTTAATGAACATTCCAAGGCC |
| ROXY10 | AT5G18600 | GGAGCAAATCCAGCGGTTTACGAGC | CATGACCTCGTTGGCTCCACCGA |
| | AT2G26695 | GATGGAAAACTGGCGATTGGGTT | GCCACCATAATCTCTTGTTGTTCTTGCATT |
| | AT1G67865 | TTGGGGTTGCCGGAGGGATTACC | ACCATTGCCGTAGTTTCCCGGACTC |
| NRT2.4 | AT5G60770 | CCGTCTTCTCCATGTCTTTC | CTGACCATTGAACATTGTGC |
| GDH3 | AT3G03910 | GCAGCTCTAGGGGGAGTCAT | CAGCCTCAGGATCAGTTGGG |

into pFLASH, the binary GATEWAY vector for promoter LUC assay (*Schultz et al., 2001*). To generate effectors, the coding region of *NIGT1.2*, *NIGT1.4*, and *GFP* in pENTR/D-TOPO were integrated into pB7WG2 binary vector.

The promoter LUC assay in *N. benthamiana* leaf was conducted as previously described with slight modifications (*Kubota et al., 2017*). *Agrobacterium* carrying binary vectors was cultured in LB media containing 20 µM acetosyringone and antibiotics overnight and subsequently precipitated at 4000 rpm at room temperature for 15 min. The *Agrobacterium* pellet was resuspended using inoculation buffer (10 mM $MgCl_2$, 10 mM MES-KOH pH 5.6, and 100 µM acetosyringone). After incubation for 3 hr at room temperature, $OD_{600}$ was adjusted to 0.2 and infiltrated into the abaxial side of tobacco leaves using a 1 mL syringe. *Agrobacterium* carrying reporters (*pFT:LUC* and *pFTm:LUC*), effector (*p35S:NIGT1.2*, *p35S:NIGT1.4* and *p35S:GFP* in the binary vector pB7WG2) were co-inoculated with pBIN61 P19 (*Voinnet et al., 2003*). Two days after inoculation, leaf discs were punched out and their LUC activities were measured using Dual-Luciferase Reporter Assay System kit (Promega, Madison, WI) and SpectraMax M5e (Molecular Devices, San Jose, CA).

## Material availability statement

All plasmids and *Arabidopsis* transgenic lines newly created in this study are available upon request to the corresponding author.

## Acknowledgements

We thank NM Belliveau and DW Galbraith for technical advice for FACS, Y Mizuta for providing research materials, M Ashikari, H Tsuji, K Nagai, and M Mizutani for scientific discussion. We also thank T Niwa, A Furuta, S Ishikawa and T Takagi for technical assistance. This research was supported by grants from the National Institutes of Health grant (R01GM079712 to J T C, C Q, and T I; NIGMS MIRA grant no. 1R35GM139532 to C Q), from the National Science Foundation (PlantSynBio grant no. 2240888 to C Q; NSF-IOS #1755452 to J L P -P), and JSPS KAKENHI (JP20H05910, JP22H04978 to T I and 24K18139 to H T).

## Additional information

### Funding

| Funder | Grant reference number | Author |
|---|---|---|
| National Institute of General Medical Sciences | R01GM079712 | Christine Quietsch<br>Josh T Cuperus<br>Takato Imaizumi |
| National Institute of General Medical Sciences | R35GM139532 | Christine Quietsch |
| National Science Foundation | 2240888 | Christine Quietsch |
| Ministry of Education, Culture, Sports, Science and Technology | JP20H05910 | Takato Imaizumi |
| Ministry of Education, Culture, Sports, Science and Technology | JP22H04978 | Takato Imaizumi |
| Japan Society for the Promotion of Science | 24K18139 | Hiroshi Takagi |
| National Science Foundation | 1755452 | Jose L Pruneda-Paz |

The funders had no role in study design, data collection and interpretation, or the decision to submit the work for publication.

## Author contributions
Hiroshi Takagi, Conceptualization, Data curation, Formal analysis, Funding acquisition, Validation, Investigation, Visualization, Methodology, Writing – original draft, Project administration, Writing – review and editing; Shogo Ito, Resources, Investigation, Writing – original draft; Jae Sung Shim, Akane Kubota, Nayoung Lee, Investigation, Writing – original draft; Andrew K Hempton, Validation, Investigation, Writing – original draft; Takamasa Suzuki, Data curation, Formal analysis, Investigation; Jared S Wong, Chansie Yang, Christine T Nolan, Investigation; Kerry L Bubb, Formal analysis; Cristina M Alexandre, Methodology; Daisuke Kurihara, Takatoshi Kiba, Resources; Yoshikatsu Sato, Yasuomi Tada, Supervision; Jose L Pruneda-Paz, Resources, Formal analysis, Supervision, Funding acquisition, Investigation; Christine Quietsch, Supervision, Funding acquisition, Writing – original draft, Project administration, Writing – review and editing; Josh T Cuperus, Data curation, Formal analysis, Funding acquisition, Writing – original draft; Takato Imaizumi, Conceptualization, Supervision, Funding acquisition, Validation, Investigation, Writing – original draft, Project administration, Writing – review and editing

## Author ORCIDs
Hiroshi Takagi ⓘ https://orcid.org/0000-0003-0321-0284
Nayoung Lee ⓘ https://orcid.org/0000-0002-0353-6455
Daisuke Kurihara ⓘ https://orcid.org/0000-0003-2703-0405
Yoshikatsu Sato ⓘ https://orcid.org/0000-0002-4967-2698
Takatoshi Kiba ⓘ https://orcid.org/0000-0001-8651-0404
Takato Imaizumi ⓘ https://orcid.org/0000-0001-9396-4412

Reviewer #1 (Public review): https://doi.org/10.7554/eLife.102529.3.sa1
Reviewer #2 (Public review): https://doi.org/10.7554/eLife.102529.3.sa2
Author response https://doi.org/10.7554/eLife.102529.3.sa3

# Additional files

## Supplementary files
MDAR checklist

## Data availability
Bulk RNA-seq from sorted nuclei can be found at the NCBI short read archive bio project PRJNA1098062. snRNA-seq can be found at NCBI GEO under GSE273032. The RNA-seq data of pSUC2:NIGT1.2 and pSUC2:NIGT1.4 plants have been deposited in the DNA Data Bank of Japan (DDBJ) Sequence Read Archive under PRJDB17784.

The following datasets were generated:

| Author(s) | Year | Dataset title | Dataset URL | Database and Identifier |
|---|---|---|---|---|
| Takagi H, Ito S, Shim J, Kubota A, Hempton A, Lee N, Suzuki T, Yang C, Nolan C, Alexandre CM, Kurihara D, Sato Y, Tada Y, Kiba T, Pruneda-Paz JL, Cuperus JT, Queitsch C, Imaizumi T | 2025 | A florigen-expressing subpopulation of phloem companion cells expresses other small proteins and reveals a nitrogen-sensitive FT repressor | https://www.ncbi.nlm.nih.gov/geo/query/acc.cgi?acc=GSE273032 | NCBI Gene Expression Omnibus, GSE273032 |
| Cuperus JT, Takagi H, Imaizumi T | 2024 | Unique characteristics of phloem companion cells producing florigen at the single-cell level | https://www.ncbi.nlm.nih.gov/bioproject/?term=PRJNA1098062 | NCBI BioProject, PRJNA1098062 |
| Suzuki T, Takagi H, Imaizumi T | 2025 | The effect of NIGT1.2 and NIGT1.4 overexpression on Arabidopsis gene expression | https://ddbj.nig.ac.jp/search/entry/bioproject/PRJDB17784 | DNA Data Bank of Japan (DDBJ) Sequence Read Archive, PRJDB17784 |

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
