## [Editor Report · eLife Assessment]

This **fundamental** study uncovers the unique molecular features of Arabidopsis phloem companion cells that highly express *FLOWERING LOCUS T* (*FT*). These *FT*-expressing cells constitute a distinct subpopulation marked by elevated ATP biosynthesis and co-expression of small mobile proteins such as FLP1 and BFT, highlighting a fine balance between florigen and anti-florigen signals. Motif analyses and transgenic studies further identify NIGT1 transcription factors as direct, nitrogen-inducible repressors of *FT*, providing a mechanism for delayed flowering under nitrogen-rich conditions. Together, the **compelling** findings show that florigen-producing companion cells integrate energy metabolism, systemic protein signals, and nutrient-responsive repression to fine-tune the seasonal and nutritional regulation of flowering.

---

## [Referee Report · Reviewer #1 (Public review)]

Summary:

The authors revealed the cellular heterogeneity of companion cells (CCs) and demonstrated that the florigen gene FT is highly expressed in a specific subpopulation of these CCs in Arabidopsis. Through a thorough characterization of this subpopulation, they further identified NITRATE-INDUCIBLE GARP-TYPE TRANSCRIPTIONAL REPRESSOR 1 (NIGT1)-like transcription factors as potential new regulators of FT. Overall, these findings are intriguing and valuable, contributing significantly to our understanding of florigen and the photoperiodic flowering pathway. However, there is still room for improvement in the quality of the data and the depth of the analysis. I have several comments that may be beneficial for the authors.

Strengths:

The usage of snRNA-seq to characterize the FT-expressing companion cells (CCs) is very interesting and important. Two findings are novel: (1) Expression of FT in CCs is not uniform. Only a subcluster of CCs exhibits high expression level of FT. (2) Based on consensus binding motifs enriched in this subcluster, they further identify NITRATE-INDUCIBLE GARP-TYPE TRANSCRIPTIONAL REPRESSOR 1 (NIGT1)-like transcription factors as potential new regulators of FT.

Weaknesses:

(1) Title: "A florigen-expressing subpopulation of companion cells". It is a bit misleading. The conclusion here is that only a subset of companion cells exhibit high expression of FT, but this does not imply that other companion cells do not express it at all.

(2) Data quality: Authors opted for fluorescence-activated nuclei sorting (FANS) instead of traditional cell sorting method. What is the rationale behind this decision? Readers may wonder, especially given that RNA abundance in single nuclei is generally lower than that in single cells. This concern also applies to snRNA-seq data. Specifically, the number of genes captured was quite low, with a median of only 149 genes per nucleus. Additionally, the total number of nuclei analyzed was limited (1,173 for the pFT:NTF and 3,650 for the pSUC2:NTF). These factors suggest that the quality of the snRNA-seq data presented in this study is quite low. In this context, it becomes challenging for the reviewer to accurately assess whether this will impact the subsequent conclusions of the paper. Would it be possible to repeat this experiment and get more nuclei?

(3) Another disappointment is that the authors did not utilize reporter genes to identify the specific locations of the FT-high expressing cells (cluster 7 cells) within the CC population in vivo. Are there any discernible patterns that can be observed?

(4) The final disappointment is that the authors only compared FT expression between the nigtQ mutants and the wild type. Does this imply that the mutant does not have a flowering time defect particularly under high nitrogen conditions?

Comments on revisions:

I think the authors took my comments seriously and addressed most of my concerns. Overall, I find this to be a very interesting paper.

---

## [Referee Report · Reviewer #2 (Public review)]

This manuscript submitted by Takagi et al. details the molecular characterization of the FT-expressing cell at a single-cell level. The authors examined what genes are expressed specifically in FT-expressing cells and other phloem companion cells by exploiting bulk nuclei and single-nuclei RNA-seq and transgenic analysis. The authors found the unique expression profile of FT-expressing cells at a single-cell level and identified new transcriptional repressors of FT such as NIGT1.2 and NIGT1.4.

Although previous researchers have known that FT is expressed in phloem companion cells, they have tended to neglect the molecular characterization of the FT-expressing phloem companion cells. To understand how FT, which is expressed in tiny amounts in phloem companion cells that make up a very small portion of the leaf, can be a key molecule in the regulation of the critical developmental step of floral transition, it is important to understand the molecular features of FT-expressing cells in detail. In this regard, this manuscript provides insight into the understanding of detailed molecular characteristics of the FT-expressing cell. This endeavor will contribute to the research field of flowering time.

During the initial review process, I proposed the following two points for improving this manuscript:

(1) The most noble finding of this manuscript is the identification of NTGI1.2 as the upstream regulator of FT-expressing cluster 7 gene expression. The flowering phenotypes of the nigtQ mutant and the transgenic plants in which NIGT1.2 was expressed under the SUC2 gene promoter support that NIGT1.2 functions as a floral repressor upstream of the FT gene. Nevertheless, the expression patterns of NIGT1.2 genes do not appear to have much overlap with those of NIGT1.2-downstream genes in the cluster 7 (Figs S14 and F3). An explanation for this should be provided in the discussion section.

(2) To investigate gene expression in the nuclei of specific cell populations, the authors generated transgenic plants expressing a fusion gene encoding a Nuclear Targeting Fusion protein (NTF) under the control of various cell type-specific promoters. Since the public audience would not know about NTF without reading reference 16, some explanation of NTF is necessary in the manuscript. Please provide a schematic of the constructs the authors used to make the transformants.

The revised manuscript has addressed my comments well. I am deeply grateful for the authors' efforts to address concerns raised by me and other reviewers.

I have no doubt that the manuscript in its current form is worthy of publication in this journal and will provide valuable insights into flowering time for many readers.

---

## [Author Response]

The following is the authors’ response to the original reviews.

**Reviewer #1 (Public review):**
Summary:The authors revealed the cellular heterogeneity of companion cells (CCs) and demonstrated that the florigen gene *FT* is highly expressed in a specific subpopulation of these CCs in Arabidopsis. Through a thorough characterization of this subpopulation, they further identified NITRATE-INDUCIBLE GARP-TYPE TRANSCRIPTIONAL REPRESSOR 1 (NIGT1)-like transcription factors as potential new regulators of FT. Overall, these findings are intriguing and valuable, contributing significantly to our understanding of florigen and the photoperiodic flowering pathway. However, there is still room for improvement in the quality of the data and the depth of the analysis. I have several comments that may be beneficial for the authors.Strengths:The usage of snRNA-seq to characterize the *FT*-expressing companion cells (CCs) is very interesting and important. Two findings are novel: (1) Expression of *FT* in CCs is not uniform. Only a subcluster of CCs exhibits high expression level of *FT*. (2) Based on consensus binding motifs enriched in this subcluster, they further identify NITRATE-INDUCIBLE GARP-TYPE TRANSCRIPTIONAL REPRESSOR 1 (NIGT1)-like transcription factors as potential new regulators of FT.

We are pleased to hear that reviewer 1 noted the novelty and importance of our work. As reviewer 1 mentioned, we are also excited about the identification of a subcluster of companion cells with very high FT expression. We believe that this work is an initial step to describe the molecular characteristics of these FT-expressing cells. We are also excited to share our new findings on NIGT1s as potential FT regulators. We believe this finding will attract a broader audience, as the molecular factor coordinating plant nutrition status with flowering time remains largely unknown despite its well-known phenomenon.

Weaknesses:(1) Title: "A florigen-expressing subpopulation of companion cells". It is a bit misleading. The conclusion here is that only a subset of companion cells exhibit high expression of FT, but this does not imply that other companion cells do not express it at all.

We agree with this comment, as it was not our intention to sound like that FT is not produced in other companion cells than the subpopulation we identified. We revised the title to more accurately reflect the point. The new title is “Companion cells with high florigen production express other small proteins and reveal a nitrogen-sensitive FT repressor.”

(2) Data quality: Authors opted for fluorescence-activated nuclei sorting (FANS) instead of traditional cell sorting method. What is the rationale behind this decision? Readers may wonder, especially given that RNA abundance in single nuclei is generally lower than that in single cells. This concern also applies to snRNA-seq data. Specifically, the number of genes captured was quite low, with a median of only 149 genes per nucleus. Additionally, the total number of nuclei analyzed was limited (1,173 for the pFT:NTF and 3,650 for the pSUC2:NTF). These factors suggest that the quality of the snRNA-seq data presented in this study is quite low. In this context, it becomes challenging for the reviewer to accurately assess whether this will impact the subsequent conclusions of the paper. Would it be possible to repeat this experiment and get more nuclei?

We appreciate this comment; we noticed that we did not clearly explain the rationale for using single-nucleus RNA sequencing (snRNA-seq) instead of single-cell RNA-seq (scRNA-seq). As reviewer 1 mentioned, RNA abundance in scRNA-seq is higher than in snRNA-seq. To conduct scRNA-seq using plant cells, protoplasting is the necessary step. However, in our study, protoplasting has many drawbacks in isolating our target cells from the phloem. First, it is technically challenging to efficiently isolate protoplasts from highly embedded phloem companion cells from plant tissues. Typically, at least several hours of enzymatic incubation are required to obtain protoplasts from companion cells (often using semi-isolated vasculatures), and the efficiency of protoplasting vasculature cells remains low. Secondly, for our analysis, restoring the time information within a day is also crucial. Therefore, we employed a more rapid isolation method. In the revision, we will explain our rationale for choosing snRNA-seq due to the technical limitations. In the revised manuscripts, we added four new sentences in the Introduction section to clearly explain these points.

Reviewer 1 also raised a concern about the quality of our snRNA-seq data, referring to the relatively low readcounts per nucleus. Although we believe that shallow reads do not necessarily indicate low quality and are confident in the accuracy of our snRNA-seq data, as supported by the detailed follow-up experiments (e.g., imaging analysis in Fig. 4B), we agree that it is important to address this point in the revision and alleviate readers’ concerns regarding the data quality.

We believe the primary reason for the low readcounts per cell is the small amount of RNA present in each Arabidopsis vascular cell nucleus that we isolated. For bulk nuclei RNAseq, we collected 15,000 nuclei. However, the total RNA amount was approximately 3 ng. It indicates that each nucleus isolated contains a very limited amount of RNA (by the simple calculation, 3,000 pg / 15,000 nuclei = 0.2 pg/nucleus). It appears that the size of cells and nuclei was still small in 2-week-old seedlings; thus, each nucleus may contain lower levels of RNA. During the optimization process, we also tried to fix the tissues that we hoped to restore nuclear retained RNA, but unfortunately, in our hands, we encountered the technical issue of nuclei aggregation that hindered the sorting process, which is not suitable for single-nucleus RNA-seq.

Reviewer 1 suggested that we repeat the same snRNA-seq experiment. We agree that having more cells increases the reliability of data. However, to our knowledge, higher cell numbers enhance the confidence of clustering, but not readcounts per cell. In our snRNAseq data, our target, FT-expressing cells, were observed in cluster 7, which projected at an obvious distance from other cell clusters. Therefore, we think that having more nuclei does not significantly help in separating high FT-expressing cluster 7 cells and different types of cells, although we may obtain more DEGs from the cluster 7 cells. Considering the costs and time required for additional snRNA-seq experiments, we think that adding more followup molecular biology experiment data would be more practical. We clearly stated the limitations of our approach in the Discussion section. “A drawback of our snRNA-seq analysis was shallow reads per nucleus. It appears mainly due to the low abundance of mRNA in nuclei from 2-week-old leaves. Based on our calculation, the average mRNA level per nucleus is approximately 0.2 pg (3,000 pg mRNA from 15,000 sorted nuclei). Future technological advance is needed to improve the data quality“

In this revised version of the manuscript, we silenced FT gene expression using an amiRNA against FT driven by tissue-specific promoters [pROXY10, cluster 7; pSUC2, companion cells; pPIP2.6, cluster 4 (for the spatial expression pattern of PIP2.6, please see the new data shown in Fig. S8F); pGC1, guard cells]. Given that both FT and ROXY10 were highly expressed in cluster 7 of our snRNA-seq dataset, we anticipated the late flowering phenotype of pROXY10:amiRNA-ft. As we expected, pROXY10:amiR-ft but not pPIP2.6:amiR-ft lines showed delayed flowering phenotypes (Fig. S14A), supporting the validity of our snRNA-seq approach. We are also now more confident in the resolution of our snRNA-seq analysis, since cluster 4-specific PIP2.6 did not cause late flowering despite its higher basal expression than ROXY10 (Fig. S14B).

(3) Another disappointment is that the authors did not utilize reporter genes to identify the specific locations of the FT-high expressing cells (cluster 7 cells) within the CC population in vivo. Are there any discernible patterns that can be observed?

In the original manuscript, as we showed only limited spatial images of overlap between FT and other cluster 7 genes in Fig. 4B, this comment is totally understandable. To respond to it, we added whole leaf images showing the spatial expression of FT and other cluster 7 genes (Fig. S12). These data indicate that cluster 7 genes including FT are expressed highly in minor veins in the distal part of the leaf but weakly in the main vein. We also added enlarged images of spatial expression of FT and cluster 7 genes (FLP1 and ROXY10) to note that those genes do not overlap completely (Fig. S13).

In contrast to cluster 7 genes, genes highly expressed in cluster 4, such as LTP1 and MLP28, are reportedly highly expressed in the main leaf vein. To further confirm it, we established a transgenic line that expresses a GFP-fusion protein controlled by the promoter of a cluster 4-specific gene PIP2.6 (Fig. S8F). It also showed strong GFP signals in the main vein, consistent with previous observations of LTP1 and MLP28. In summary, FT-expressing cells (cluster 7 cells) are enriched in companion cells in the minor vein, and their expression patterns show a clear distinction from genes expressed in the main vein (e.g., cluster 4-specific genes).

(4) The final disappointment is that the authors only compared FT expression between the nigtQ mutants and the wild type. Does this imply that the mutant does not have a flowering time defect particularly under high nitrogen conditions?

We agree with reviewer 1 that more experiments are required to conclude the role of NIGT1 on FT regulation, in addition to our Y1H data, flowering time data of NIGT1 overexpressors, and FT expression in NIGT1 overexpressors and nigtQ mutant.

First, to test the direct regulation of NIGT1s on FT transcription, we conducted a transient luciferase (LUC) assay in tobacco leaves using effectors (p35S:NIGT1.2, p35S:NIGT1.4, and p35S:GFP) and reporters [pFT:LUC (FT promoter fused with LUC) and pFTm:LUC (the same FT promoter with mutations in NIGT1-binding sites fused with LUC)]. Our result showed that NIGT1.2 and NIGT1.4, but not GFP, decreased the activity of pFT:LUC but not pFTm:LUC (Fig. 5C). This indicates that NIGT1s directly repress the FT gene.

Second, to address reviewer 1’s suggestion about the effect of of nigtQ mutation on flowering time, we have grown WT and nigtQ plants on 20 mM and 2 mM NH_4_NO_3_. Under 20 mM NH_4_NO_3_, the nigtQ line bolted at earlier days than WT; under 2 mM NH_4_NO_3_, nigtQ and WT bolted at almost same timing (Fig. S17D and E). This result suggests that the nigtQ mutation affects flowering timing depending on nitrogen nutrient status. However, leaf numbers of bolted plants were not different between WT and nigtQ lines (Fig. S17E). Therefore, it appears that nigtQ mutation also accelerated overall growth of plants rather than flowering promotion. We also have measured flowering time by counting leaf numbers of the nigtQ and WT plants at bolting on nitrogen-rich soil. The mutant generated slightly more leaves than WT when they flowered (Fig. S17G). These results suggest that the NIGT-derived fine-tuning of FT regulation is conditional on higher nitrogen conditions.

Minor:(1) Abstract: "Our bulk nuclei RNA-seq demonstrated that FT-expressing cells in cotyledons and in true leaves differed transcriptionally.". This sentence is not informative. What exactly is the difference in FT-expressing cells between cotyledons and true leaves?

We modified the sentence to clarify the differences between cotyledons and true leaves. “Our bulk nuclei RNA-seq demonstrated that FT-expressing cells in cotyledons and true leaves showed differences especially in FT repressor genes.”

(2) As a standard practice, to support the direct regulation of FT by NIGT1, the authors should provide EMSA and ChIP-seq data. Ideally, they should also generate promoter constructs with deletions or mutations in the NIGT1 binding sites.

To test direct interaction of NIGT1 to the FT promoter sequences, we performed the transient reporter assay using FT promoter driven luciferase reporter (Fig. 5C). NIGT1.2 and NIGT1.4 repressed the FT promoter activity; however, with NIGT1 binding site mutations, this repression was not observed, indicating that NIGT1 binds to the ciselements in the FT promoter to repress its transcription.

(3) Sorting: Did the authors fix the samples before preparing the nuclei suspension? If not, could this be the reason the authors observed the JA-responsive clusters (Fig. 2J)? Please provide more details related to nuclei sorting in the Methods section.

We added a new subsection in the Materials and Methods section to explain a detail of the nuclei sorting procedure. We did not include a sample fixation step. We have tried formaldehyde fixation; however, it clumped nuclei, which was not suitable for snRNA-seq. Moreover, fixation steps generally reduce readcounts of single-cell RNA-seq according to the 10X Genomics’ guideline.

We agree that JA responses were triggered during the FANS nuclei isolation. Therefore, we added the following sentence. “Since our FANS protocol did not include a sample fixation step to avoid clumping, these cells likely triggered wounding responses during the chopping and sorting process (Fig. S1B).

**Reviewer #2 (Public review):**
This manuscript submitted by Takagi et al. details the molecular characterization of the FTexpressing cell at a single-cell level. The authors examined what genes are expressed specifically in FT-expressing cells and other phloem companion cells by exploiting bulk nuclei and single-nuclei RNA-seq and transgenic analysis. The authors found the unique expression profile of FT-expressing cells at a single-cell level and identified new transcriptional repressors of FT such as NIGT1.2 and NIGT1.4.Although previous researchers have known that FT is expressed in phloem companion cells, they have tended to neglect the molecular characterization of the FT-expressing phloem companion cells. To understand how FT, which is expressed in tiny amounts in phloem companion cells that make up a very small portion of the leaf, can be a key molecule in the regulation of the critical developmental step of floral transition, it is important to understand the molecular features of FT-expressing cells in detail. In this regard, this manuscript provides insight into the understanding of detailed molecular characteristics of the FT-expressing cell. This endeavor will contribute to the research field of flowering time.We are grateful that reviewer 2 recognizes the importance of transcriptome profiling of FTexpressing cells at the single-cell level.Here are my comments on how to improve this manuscript.(1) The most noble finding of this manuscript is the identification of NTGI1.2 as the upstream regulator of FT-expressing cluster 7 gene expression. The flowering phenotypes of the nigtQ mutant and the transgenic plants in which NIGT1.2 was expressed under the SUC2 gene promoter support that NIGT1.2 functions as a floral repressor upstream of the FT gene. Nevertheless, the expression patterns of NIGT1.2 genes do not appear to have much overlap with those of NIGT1.2-downstream genes in the cluster 7 (Figs S14 and F3). An explanation for this should be provided in the discussion section.

We agree with reviewer 2 that the spatial expression patterns of NIGT1.2 and cluster 7 genes do not overlap much, and some discussion should be provided in the manuscript. Although we do not have a concrete answer for this phenomenon, we obtained the new data showing that NIGT1.2 and NIGT1.4 directly repress the FT gene in planta (Fig. 5C). As NIGT1.2/1.4 are negative regulators of FT, it is plausible that NIGT1.2/1.4 may suppress FT gene expression in non-cluster 7 cells to prevent the misexpression of FT. We added this point in the Results section.

(2) To investigate gene expression in the nuclei of specific cell populations, the authors generated transgenic plants expressing a fusion gene encoding a Nuclear Targeting Fusion protein (NTF) under the control of various cell type-specific promoters. Since the public audience would not know about NTF without reading reference 16, some explanation of NTF is necessary in the manuscript. Please provide a schematic of constructs the authors used to make the transformants.

As reviewer 2 pointed out, we lacked a clear explanation of why we used NTF in this study. NTF is the fusion protein that consists of a nuclear envelope targeting WPP domain, GFP, and a biotin acceptor peptide. It was initially designed for the INTACT (isolation of nuclei tagged in specific cell types) method, which enables us to isolate bulk nuclei from specific tissues. Although our original intention was to profile the bulk transcriptome of mRNAs that exist in nuclei of the FT-expressing cells using INTACT, we utilized our NTF transgenic lines for snRNA-seq analysis. To explain what NTF is to readers, we included a schematic diagram of NTF (Fig. S1A) and more explanation about NTF in the Results section.

Again, we appreciate all reviewers’ careful and constructive comments. With these changes, we hope our revised manuscript is now satisfactory.